# General Transportability of Soft Interventions: Completeness Results

**Juan D. Correa**          **Elias Bareinboim**
Causal Artificial Intelligence Laboratory
Columbia University
{jdcorrea,eb}@cs.columbia.edu

## Abstract

The challenge of generalizing causal knowledge across different environments is pervasive in scientific explorations, including in AI, ML, and Data Science. Experiments are usually performed in one environment (e.g., in a lab, on Earth) with the intent, almost invariably, of being used elsewhere (e.g., outside the lab, on Mars), where the conditions are likely to be different. In the causal inference literature, this generalization task has been formalized under the rubric of *transportability* (Pearl and Bareinboim, 2011), where a number of criteria and algorithms have been developed for various settings. Despite the generality of such results, transportability theory has been confined to atomic, *do()*-interventions. In practice, many real-world applications require more complex, stochastic interventions; for instance, in reinforcement learning, agents need to continuously adapt to the changing conditions of an uncertain and unknown environment. In this paper, we extend transportability theory to encompass these more complex types of interventions, which are known as "soft," both relative to the input as well as the target distribution of the analysis. Specifically, we develop a graphical condition that is both necessary and sufficient for deciding soft-transportability. Second, we develop an algorithm to determine whether a non-atomic intervention is computable from a combination of the distributions available across domains. As a corollary, we show that the $\sigma$-calculus is complete for the task of soft-transportability.

## 1   Introduction

Generalizing causal knowledge across disparate domains (i.e., populations, settings, environments) is at the heart of many inferences across the empirical sciences as well as AI [26, 33, 29]. In economics, for example, the Nobel Prize of 2019 was awarded to Duflo, Banerjee, and Kremer "for their experimental approach to alleviating global poverty". Their work is in fact about systematically performing experiments on the effect of complex policies related to poverty, and then carefully trying to extrapolate the gathered evidence to other populations [1, 12].[1] The same need to generalize across disparate conditions is present in Reinforcement Learning. For instance, consider a rover trained in the California desert for digging rocks. After exhaustive months of training, NASA wants to deploy the vehicle on Mars, where the environment is not exactly the same, but somewhat similar to Earth. The expectation is that the rover will need minimal "experimentation" (i.e., trial-and-error) on Mars by leveraging the knowledge acquired here, operating more surgically and effectively there.

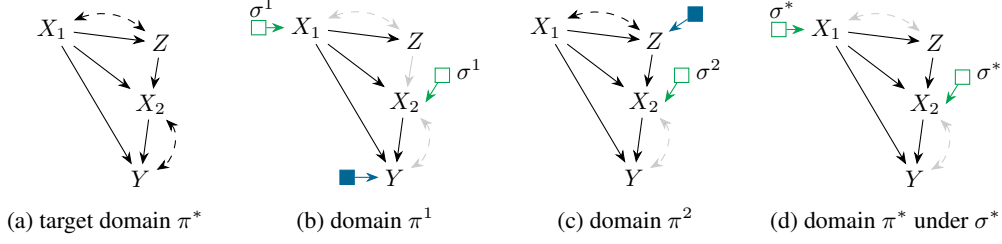

(a) target domain $\pi^*$      (b) domain $\pi^1$      (c) domain $\pi^2$      (d) domain $\pi^*$ under $\sigma^*$

Figure 1: (a) A causal graph representing the target domain $\pi^*$ (before intervention). (b) and (c) filled squared nodes encode the differences between $\pi^*$ and $\pi^1, \pi^2$. Squared hollow nodes indicate intervened variables due to policies $\sigma^1, \sigma^2$. (d) represents $\pi^*$ under the intended policy $\sigma^*_{X_1,X_2}$.

The solution to these apparently disparate tasks faced by Duflo, NASA, and so many other scientists, perhaps surprisingly, relies on a common ingredient, namely, the invariances of some causal mechanisms shared across the different settings. The task of leveraging causal invariances so as to extrapolate experimental knowledge across settings has been formally studied in the causal inference literature under the rubric of *transportability* [28, 2, 19, 20, 4, 5]. Recent work by [22] provided complete graphical and algorithmic treatment for this task, bringing under the same umbrella previous identification results on how to combine observations and experiments [36, 34, 31, 15, 3, 21].

Most of this literature, however, focuses on *atomic* interventions, often times represented by the celebrated *do*-operator [25, 26, Ch. 3]. Formally, $do(X{=}x)$ represents the symbolic operation of replacing the underlying causal mechanism that naturally dictates the behavior of variable $X$ with a constant value $x$. While the $do(\cdot)$ is a grounding element in causal analysis, many real-world situations involve *soft interventions*, or *policy interventions*, which respond in a stochastic manner to other variables in the system [38, 14, 7, 27]. The do-operator was used to support some types of soft interventions [26, Ch. 4], and was expanded towards a broader set of interventions in the context of dynamic plans [30, 18, 13, 35, 11, 32, 8, 9]. Despite the generality of these results, there is still no formal treatment of transportability/generalizability settings in the context of soft interventions.

For concreteness, consider a medical setting (e.g., in the context of HIV, Cancer, and Chronic Diseases [23, 24]), where patient data collected under different treatment protocols and locations need to be combined to generate a policy in a different site. In the causal graph in Fig. 1(a), $X_1$ and $X_2$ are treatments applied in sequence, $Z$ is a secondary condition detected after the first treatment ($X_1$) that affects the application of $X_2$, and $Y$ represents survival. With the task of designing a new protocol $\sigma^*_{X_1,X_2}$ to treat the condition, three sources of data are available:

1. an observational study from the current hospital ($\pi^*$), where the protocol is to be implemented;
2. a controlled study from hospital $\pi^1$ that is under policy $\sigma^1$, where treatment $X_1$ is randomized and $X_2$ administered only if $X_1$ is given; and
3. another controlled study from hospital $\pi^2$ that follows protocol $\sigma^2$, where the treatment $X_1$ is applied as in $\pi^*$ but $X_2$ is determined by $X_1$ and the secondary condition $Z$.

Figs. 1(b),(c) represent the *selection diagrams* (to be formally defined in the next section) that show the qualitative differences between $\pi^*$ and, respectively, $\pi^1$ and $\pi^2$. The square nodes pointing to $Y$ and $Z$ indicate that the corresponding distributions are different for the populations of hospitals $\pi^1$ and $\pi^2$, respectively. The square nodes annotated with $\sigma^1$ and $\sigma^2$ point to treatment variables following the protocols described above. Given the causal description of these protocols, the question is then – how to combine these various datasets, collected across disparate conditions, so as to estimate the effect of the new policy in $\pi^*$, i.e., $P^*(Y; \sigma^*_{X_1,X_2})$? It turns out that the new policy can be evaluated from the available data using the following formula:

$$P^*(y; \sigma^*_{X_1,X_2}) = \sum_{x_1,z,x_2} \underbrace{P^2(y \mid x_1, x_2; \sigma^2)}_{\text{from } \pi^2} \underbrace{P^1(z \mid x_1; \sigma^1)}_{\text{from } \pi^1} \underbrace{P(x_2 \mid z, x_1; \sigma^*)P(x_1; \sigma^*)}_{\text{defined by } \sigma^*_{X_1,X_2}} \quad (1)$$

This is a delicate mixture of soft experimental data from $\pi^1$ and $\pi^2$, where each of the factors alone are not sufficient, but combined they are necessary for the evaluation of the new policy. In this paper, our goal is to explicate this non-trivial formula, and, more broadly, characterize under what conditions the causal effect of a new policy can be computed from disparate data sources.

| Type | Strategy | Function |
|---|---|---|
| Idle | $\sigma_X{=}\emptyset$ | $f'_x = f_x$ |
| Atomic/*do* | $\sigma_X{=}x$ | $f'_x = x$, for some $x \in Dom(X)$ |
| Conditional | $\sigma_X{=}g(\mathbf{Pa}'_x)$ | $f'_x = g(\mathbf{Pa}'_x)$ |
| Stochastic/Random | $\sigma_X{=}P'(X|\mathbf{Pa}'_x)$ | $f'_x$ s.t. $P'(x|\mathbf{Pa}'_x){=}\sum_{\mathbf{u}'_x} P(f'_x(\mathbf{Pa}'_x, \mathbf{u}_x){=}x)P(\mathbf{u}_x)$ |

Table 1: Summary of the types of interventions considered. Each row contains the type of intervention, its representation using the regime indicator and the way the corresponding replacement function.

More specifically, our contributions are as follow:

1. **Reduction to atomic case.** We reduce the problem of transporting $P^*(\mathbf{y}; \sigma_\mathbf{X})$ to that of transporting the effect of an atomic-intervention. We then prove the tightness of the reduction.
2. **Algorithmic solution.** We design an efficient algorithm to determine the existence of an estimand for the effect of a non-atomic intervention as a function of the available distributions.
3. **Symbolic characterization.** We prove that the $\sigma$-calculus is necessary and sufficient for the task of transportability when both the input and the output distributions involve soft interventions.
4. **Graphical characterization.** We derive a complete graphical condition for this task.

**Preliminaries.** Random variables are represented with uppercase letters (e.g, $C$) and their realizations with lowercase ones (e.g, $c$). Letters in bold (e.g, $\mathbf{C}$) represent sets of variables, and lowercase-bold letters (e.g., $\mathbf{c}$) a joint value assignment for the variables in the set. Given a graph $\mathcal{G}$, $\mathcal{G}_{\overline{\mathbf{W}}\underline{\mathbf{X}}}$ is the result of removing edges coming into variables in $\mathbf{W}$ and going out from variables in $\mathbf{X}$. Also, $\mathcal{G}_{[\mathbf{C}]}$ is the subgraph of $\mathcal{G}$ made only of nodes in $\mathbf{C}{\subset}\mathbf{V}$ and the edges between them. We define $Pa(\mathbf{C})$ and $An(\mathbf{C})$, as the union of $\mathbf{C}{\subset}\mathbf{V}$ with its parents and ancestors in the graph, respectively.

We use the Structural Causal Model (SCM) paradigm [26, Ch. 7] to represent the causal structure of the system and elucidate the effects of changing it. An SCM $\mathcal{M}$ is a 4-tuple $\langle \mathbf{U}, \mathbf{V}, \mathcal{F}, P(\mathbf{u}) \rangle$, where $\mathbf{U}$ is a set of exogenous (latent) variables; $\mathbf{V}$ is a set of endogenous (observable) variables; $\mathcal{F}$ is a collection of functions $\{f_i\}_{V_i \in \mathbf{V}}$. Each $f_i$ is a mapping from a set of exogenous variables $\mathbf{U}_i \subseteq \mathbf{U}$ and a set of endogenous variables $\mathbf{Pa}_i \subseteq \mathbf{V} \setminus \{V_i\}$ to the domain of $V_i$. The uncertainty is encoded through a probability distribution over the exogenous variables, $P(\mathbf{U})$. Each SCM $\mathcal{M}$ induces a *causal diagram* where every $V_i \in \mathbf{V}$ is a vertex, there is a directed edge $(V_j \to V_i)$ for every $V_i \in \mathbf{V}$ and $V_j \in \mathbf{Pa}_i$, and a bidirected edge $(V_i \leftarrow\!\text{-}\!\text{-}\!\text{-}\!\rightarrow V_j)$ for every pair $V_i, V_j \in \mathbf{V}$ such that $\mathbf{U}_i \cap \mathbf{U}_j \neq \emptyset$ ($V_i$ and $V_j$ have a common exogenous parent). We assume the underlying model is recursive, that is, there are no cyclic dependencies among the variables.

An intervention on a variable $X$ replaces $f_x$ with a new function $f'_x$ of some $\mathbf{Pa}'_x \subseteq \mathbf{V} \setminus \{X\}$ and variables $\mathbf{U}'_x$. $\mathbf{Pa}'_x$ could differ from $\mathbf{Pa}_x$, and $\mathbf{U}'_x \cap \mathbf{U} = \emptyset$. In particular, $\mathbf{Pa}'_x$ need not be a subset of $\mathbf{Pa}_x$ as long as it does not include any descendant of $X$. We consider four types of interventions summarized in Table 1. An *idle* intervention leaves the function as it is, so we often omit $\sigma_X{=}\emptyset$ in any expression. *Atomic* interventions fix the intervened variable to a constant value. *Conditional* and *stochastic* interventions allow the intervened variable to change as a deterministic function or a conditional probability distribution of a set of observable parents. For interventions on a set of variables $\mathbf{X} \subseteq \mathbf{V}$ let $\sigma_\mathbf{X}{=}\{\sigma_X\}_{X \in \mathbf{X}}$, that is, the result of applying one intervention after the other. Each $\sigma_X$ affects a different variable, so the order in which they are considered is not important. Given an intervention $\sigma_\mathbf{X}$ a new model can be defined as $\mathcal{M}_{\sigma_\mathbf{X}} = \langle \mathbf{V}, \mathbf{U} \cup \mathbf{U}'_\mathbf{x}, \mathcal{F}', P(\mathbf{U}, \mathbf{U}'_\mathbf{x}) \rangle$, where $\mathbf{U}'_\mathbf{x} = \bigcup_{X \in \mathbf{X}} \mathbf{U}'_x$ and $\mathcal{F}' = (\mathcal{F} \setminus \{f_x\}_{X \in \mathbf{x}}) \cup \{f'_x\}_{X \in \mathbf{x}}$. Hereafter, we will only consider interventions that result in a recursive $\mathcal{M}_{\sigma_\mathbf{X}}$. The model $\mathcal{M}_{\sigma_\mathbf{X}}$ induces a probability distribution

$$P(\mathbf{v}; \sigma_\mathbf{X}) = \sum_{\mathbf{u}, \mathbf{u}'_\mathbf{x}} \prod_{\{i|V_i \in \mathbf{X}\}} P(v_i|\mathbf{pa}_i, \mathbf{u}_i, \mathbf{u}'_i; \sigma_\mathbf{X}) P(\mathbf{u}'_\mathbf{x}; \sigma_\mathbf{X}) \prod_{\{i|V_i \in \mathbf{V}\setminus\mathbf{X}\}} P(v_i|\mathbf{pa}_i, \mathbf{u}_i) P(\mathbf{u}), \quad (2)$$

and a causal graph $\mathcal{G}_{\sigma_\mathbf{X}}$. Here the notation $P(\cdot\,; \sigma_\mathbf{X})$ is used to refer to the probability distribution under $\sigma_\mathbf{X}$ similar to the way $P(\cdot \mid do(\mathbf{x}))$ is used in the atomic case. Following the representation in [8] we annotate $\mathcal{G}_{\sigma_\mathbf{X}}$ with extra nodes called *regime indicators* [10] to mark the variables in $\mathcal{M}_{\sigma_\mathbf{X}}$ whose functions have been replaced relative to $\mathcal{M}$. Specifically, $\mathcal{G}_{\sigma_\mathbf{X}}$ contains a node $\sigma_{X_i}$ for every $X_i \in \mathbf{X}$ together with an edge $(\sigma_{X_i} \to X_i)$.

The proofs are provided in the Appendix (Supplemental Material).

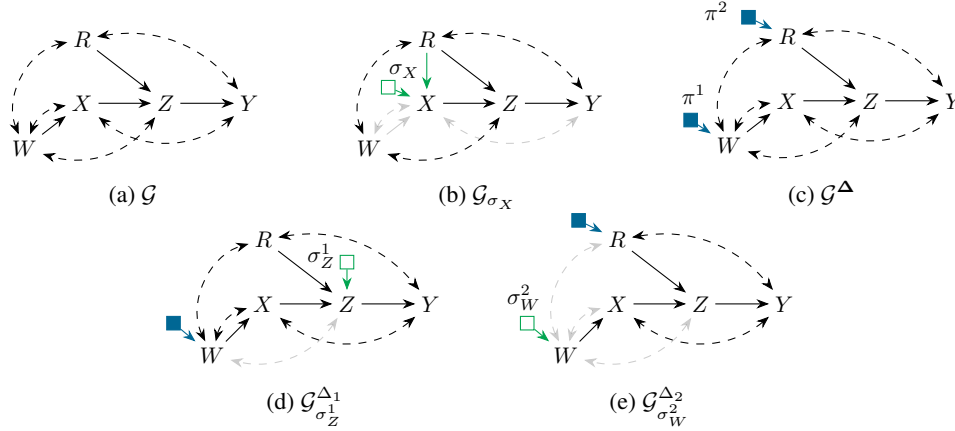

Figure 2: Diagram (a) represents the natural regime in $\pi^*$ and (b) the regime after the target intervention. The selection diagram (c) compares $\pi^*$ with $\pi^1$ and $\pi^2$, while (d) and (e) are selection diagrams specific to domains $\pi^1$ and $\pi^2$ under interventions $\sigma_Z^1$ and $\sigma_W^2$, respectively.

## 2 Transporting Causal Relations across Domains

Our goal is to predict the effect of a soft intervention $\sigma_{\mathbf{X}}$ using assumptions encoded in the form of a graph and different observational and/or experimental distributions arising from different domains. Let $\Pi = \{\pi^*, \pi^1, \pi^2, \ldots\}$ be the set of domains/populations involved in the analysis where $\pi^*$ has a special status as the *target* domain where the query is to be inferred.

To better understand this setting, let us consider an example of a hypothetical study of government-backed loan programs and successful payment of family house purchases (inspired on studies such as [6, 17]). In city $\pi^*$, the percentage of the value of the property that can be borrowed ($X$) depends mainly on the credit history of the applicant ($W$) and his current employment conditions. The amount borrowed (or principal), $Z$, depends on the allowed percentage and the characteristics ($R$) of the property — mainly cost and location — as well as financial aspects of the borrower. The goal of the policy is to increase the number of families purchasing their own homes and paying their mortgages, so $Y$ represents an indicator of the progress of the payment after a certain amount of years. Fig. 2(a) represents this situation in $\pi^*$. Let $\pi^1$ be another city where the distributions of credit history $W$ is different than in $\pi^*$, and $\pi^2$ another where the distribution of $R$ is the one that differs. These differences are called "domain discrepancies", formally defined next.

**Definition 1** (Domain Discrepancy)**.** Let $\pi^a$ and $\pi^b$ be domains associated, respectively, with SCMs $\mathcal{M}^a$ and $\mathcal{M}^b$ conforming to a causal diagram $\mathcal{G}$. We denote by $\Delta^{a,b} \subseteq \mathbf{V}$ a set of variables such that, for every $V_i \in \Delta^{a,b}$, there might exist a discrepancy if $f_i^a \neq f_i^b$ or $P^a(\mathbf{U}_i) \neq P^b(\mathbf{U}_i)$.

We will write $\Delta^{*,i}$ simply as $\Delta^i$ to represent the differences between the target and each source domain, with $\Delta^*=\emptyset$. In this example, $\Delta^1=\{W\}$ and $\Delta^2=\{R\}$, which can be encoded in a *selection diagram* $\mathcal{G}^{\mathbf{\Delta}}$ shown in Fig. 2(c), which according to the following definition:

**Definition 2** (Selection Diagram)**.** Given a causal diagram $\mathcal{G}^i = \langle \mathbf{V}, \mathbf{E} \rangle$ and domain discrepancies $\Delta^i$, let $\mathbf{S} = \{S_v \mid \exists_{i=1}^n V \in \Delta^i\}$ be called *selection variables*. Then, a *selection diagram* $\mathcal{G}^{\Delta_i}$ is defined as a graph $\langle \mathbf{V} \cup \mathbf{S}, \mathbf{E} \cup \{S_v \to V\}_{S_v \in \mathbf{S}} \rangle$.

Policy makers in city $\pi^*$ are planning to increase the percentage $X$ by offering loaners insurance from default as long as the percentage ought is above the regular threshold, for properties within certain locations (accounted in $R$). We denote this policy as $\sigma_X^*$ and represent the resulting regime in Fig. 2(b). To assess the impact of such policy, data of the current mortgage payments in $\pi^*$ have been collected together with data from other two cities, $\pi^1$ and $\pi^2$. The administration of $\pi^1$ allocated loans where the amount $Z$ was determined as a function of the allowed percentage ($X$) and the property ($R$), as depicted in Fig. 2(d). In city $\pi^2$, a study selected borrowers at random and had loaners process their applications with randomized credit history $W$, to observe its effect on $Y$ (Fig. 2(e)). Data available in each domain is specified by $\mathbb{Z} = \{\mathbb{Z}^i \mid \pi^i \in \Pi\}$, where each $\mathbb{Z}^i = \{\sigma_{\mathbf{Z}_1}, \sigma_{\mathbf{Z}_2}, \ldots\}$, $\mathbf{Z}_j \subseteq \mathbf{V}$, corresponds to domain $\pi^i$. This means that distributions

$\{P^i(\mathbf{V}; \sigma_{\mathbf{Z}_j}) \mid \mathbf{Z}_j \in \mathbb{Z}^i\}_{\mathbb{Z}^i \in \mathbb{Z}}$ are assumed to be available. Notice that $P^i(\mathbf{V}; \sigma_\emptyset) = P^i(\mathbf{V})$ is a valid distribution and describes the observational (non-interventional) distribution in domain $\pi^i$. In this example, $\mathbb{Z} = \{\mathbb{Z}^* = \{\sigma_\emptyset\}, \mathbb{Z}^1 = \{\sigma_Z\}, \mathbb{Z}^2 = \{\sigma_W\}\}$.

The effect of the new policy $\sigma_X^*$ can be measured by comparing $E[Y; \sigma_X^*]$ with $E[Y]$ for $\pi^*$. While the latter is readily estimable via $P^*(Y)$, which is part of the input; the challenge is to assess $P^*(Y; \sigma_X^*)$. The ability to infer the target effect from the input distributions is formalized next.

**Definition 3** (Effect Transportability). Let $\mathbf{Y}, \mathbf{X}, \mathbf{W} \subset \mathbf{V}$ with $\mathbf{W} \cap \mathbf{Y} = \emptyset$. The (conditional) effect intervention $\sigma_{\mathbf{X}}$ on a set of outcome variables $\mathbf{Y}$, conditional on $\mathbf{W}$, $P^*(\mathbf{y}|\mathbf{w}; \sigma_{\mathbf{X}})$, in a target environment $\pi^*$, is said to be transportable from $\langle \mathcal{G}^\Delta, \mathbb{Z} \rangle$, if it is uniquely computable from the set of distributions $\mathbb{Z}$ for every assignment $(\mathbf{y}, \mathbf{w})$ and every set of models $\{\mathcal{M}^i\}_{\pi^i \in \Pi}$ inducing $\mathcal{G}^\Delta$ and $\mathbb{Z}$.

To solve this particular transportability instance, we use $\sigma$-calculus [9] and standard probability axioms.[2] For simplicity $\sigma_X = \sigma_X^*, \sigma_Z = \sigma_Z^1$ and $\sigma_W = \sigma_W^2$ are simply written as $\sigma_X, \sigma_Z$ and $\sigma_W$.

$$P^*(y; \sigma_X) = \sum\nolimits_{r,x,z} P^*(y|z,x,r; \sigma_X)P^*(z|x,r; \sigma_X)P^*(x|r; \sigma_X)P^*(r; \sigma_X). \qquad (3)$$

By rule 3 and the separation $(R \perp\!\!\!\perp X)$ in $\mathcal{G}_{\sigma_X^* \overline{X}}$ and $\mathcal{G}_{\overline{X}}$, we have $P^*(r; \sigma_X) = P^*(r)$, which is estimable from the input distribution $P^*(\mathbf{V})$. The factor $P^*(x|r; \sigma_X)$ is determined by $\sigma_X^*$ (and the policy's specification). For the second factor, $P^*(z|x,r; \sigma_X) = P^*(z|x,r; \sigma_X, \sigma_W)$ by rule 3 and $(Z \perp\!\!\!\perp W \mid X, R)$ in $\mathcal{G}_{\sigma_X \sigma_W \overline{W}}$ and $\mathcal{G}_{\sigma_X \overline{W}}$. Then, by rule 2 and $(Z \perp\!\!\!\perp X \mid R)$ in $\mathcal{G}_{\sigma_X \sigma_W \underline{X}}$ and $\mathcal{G}_{\sigma_W \underline{X}}$, we obtain $P^*(z|x,r; \sigma_W)$. From the graph $\mathcal{G}_{\sigma_W}^{\Delta_2}$ (Fig. 2(e)), $(Z \perp\!\!\!\perp S_r \mid R, X)$ hence $P^*(z|x,r; \sigma_W) = P^2(z|x,r; \sigma_W)$ estimable from the given $P^2(\mathbf{V}; \sigma_W = \sigma_W^2)$.

The first factor is equal to $P^*(y|z,x,r; \sigma_X, \sigma_Z = z)$ by rule 2 and $(Y \perp\!\!\!\perp Z \mid X, R)$ in $\mathcal{G}_{\sigma_X \sigma_Z = z \underline{Z}}$ and $\mathcal{G}_{\sigma_X \underline{Z}}$. We remove the observed $X$ by rule 1 and $(Y \perp\!\!\!\perp X \mid Z, R)$ in $\mathcal{G}_{\sigma_X \sigma_Z = z}$. Then, by rule 3 and $(Y \perp\!\!\!\perp X \mid Z, R)$ in $\mathcal{G}_{\sigma_X \sigma_Z = z \overline{X}}$ and $\mathcal{G}_{\sigma_Z = z \overline{X}}$, this is equal to $P^*(y|z,r; \sigma_Z = z)$ and transportable from $\pi^1$ since $(Y \perp\!\!\!\perp S_w \mid Z, R)$. We sum over $X$ and use rule 2 with $(Y \perp\!\!\!\perp Z \mid X, R)$ in $\mathcal{G}_{\sigma_Z = z \underline{Z}}$ and $\mathcal{G}_{\sigma_Z \underline{Z}}$ to exchange $\sigma_Z = z$ with $\sigma_Z = \sigma_Z^1$ (see Appendix B.1 for the detailed derivation). Then

$$P^*(y; \sigma_X) = \sum_{r,x,z} \left( \underbrace{\sum\nolimits_{x'} P^1(y|z,x',r; \sigma_Z)P^1(x'|r; \sigma_Z)}_{\text{from } \sigma_Z^1 \text{ in } \pi^1} \right) \underbrace{P^2(z|x,r; \sigma_W)}_{\text{from } \sigma_W^2 \text{ in } \pi^2} \underbrace{P^*(x|r; \sigma_X^*)}_{\text{def. } \sigma_X^*} \underbrace{P^*(r)}_{\text{from } \pi^*}. \qquad (4)$$

To solve this kind of transportability instances, we can leverage the transportability theory for atomic interventions. Accordingly, we establish a crisp relationship between any instance with soft interventions and its atomic counterpart.

**Theorem 1.** *Let $\mathbf{Y}, \mathbf{X} \subseteq \mathbf{V}$ be any two sets of variables, and let $\sigma_{\mathbf{X}}^*$ be an atomic, conditional or stochastic intervention. Then, the effect of $\sigma_{\mathbf{X}}^*$ on $\mathbf{Y}$ can be written as*

$$P^*(\mathbf{y}; \sigma_{\mathbf{X}} = \sigma_{\mathbf{X}}^*) = \sum_{\mathbf{d} \setminus \mathbf{y}} P^*(\mathbf{d} \setminus \mathbf{x}; \sigma_{\mathbf{X}} = \mathbf{x}) \prod_{X \in \mathbf{X} \cap \mathbf{D}} P^*(x \mid \mathbf{pa}_x; \sigma_{\mathbf{X}} = \sigma_{\mathbf{X}}^*). \qquad (5)$$

*Moreover, this effect is transportable from $\langle \mathcal{G}^\Delta, \mathbb{Z} \rangle$ if and only if $P^*(\mathbf{d} \setminus \mathbf{x}; \sigma_{\mathbf{X}} = \mathbf{x})$ is transportable from $\langle \mathcal{G}^\Delta, \mathbb{Z} \rangle$, where $\mathbf{D} = An(\mathbf{Y})_{\mathcal{G}_{\sigma_{\mathbf{X}}}}$.*

For the example just discussed,

$$P^*(y; \sigma_X) = \sum\nolimits_{r,x,z} P^*(y,z,r; \sigma_X = x)P^*(x \mid r; \sigma_X = \sigma_X^*), \qquad (6)$$

where the first factor can be transported with a similar derivation to the one described above.

Given the tightness of the reduction provided by Thm. 1, one may surmise that it is possible to blindly use existing transportability algorithms (e.g., GTR [22]) to solve for soft interventions. In particular, if one pretends the input consist of $do()$-experiments, as expected by known algorithms, the resulting expression cannot be mapped directly in terms of the original soft-experiments (for further details, see Appendix B.3). This motivates the clean algorithmic approach undertaken in the next section.

# 3 A Complete Algorithm for Soft Transportability

Both input and output of the transportability task refer to probability distributions within different domains and for different interventions. A key building block of our algorithm is the ability to decompose such distributions in factors with the finest granularity possible. We use the concept of *C-factors* and *C-components* developed by Tian and Pearl in [36, 37]. The set $\mathbf{V}$ can be partitioned into *C-components* such that two variables belong to the same C-component if they are connected in $\mathcal{G}$ by a path made entirely of bidirected edges. For instance the graph in Fig. 1(a) induces two C-components $\{X_1, Z\}$ and $\{X_2, Y\}$.

For any $\mathbf{C} \subseteq \mathbf{V}$, the quantity $Q^k[\mathbf{C}; \sigma_{\mathbf{X}}](\mathbf{v})$ is the *C-factor* of $\mathbf{C}$ under intervention $\sigma_{\mathbf{X}}$ in domain $\pi^k$, and denotes the following function

$$Q^k[\mathbf{C}; \sigma_{\mathbf{X}}](\mathbf{v}) = \sum\nolimits_{\mathbf{u}(\mathbf{C})} \prod\nolimits_{\{i | V_i \in \mathbf{C}\}} P^k(v_i \mid \mathbf{pa}_i, \mathbf{u}_i; \sigma_{\mathbf{X}}) P^k(\mathbf{u}(\mathbf{C}); \sigma_{\mathbf{X}}), \qquad (7)$$

where $\mathbf{U}(\mathbf{C}) = \bigcup_{V_i \in \mathbf{C}} \mathbf{U}_i$. When $\mathbf{C} = \mathbf{V}$, $Q^k[\mathbf{V}; \sigma_{\mathbf{X}}](\mathbf{v}) = P^k(\mathbf{v}; \sigma_{\mathbf{X}})$. For simplicity we write $Q^k[\mathbf{C}; \sigma_{\mathbf{X}}](\mathbf{v})$ as $Q^k[\mathbf{C}; \sigma_{\mathbf{X}}]$, $Q^k[\mathbf{C}; \sigma_{\mathbf{X}}]$ as $^kQ[\mathbf{C}]$ if $\sigma_{\mathbf{X}} = \emptyset$, $Q^k[\{V_i\}; \sigma_{\mathbf{X}}]$ just as $Q^k[V_i; \sigma_{\mathbf{X}}]$, and $Q[\mathbf{C}]$ when talking about C-factors in general, for any domain and intervention.

A C-factor $Q[\mathbf{C}]$ can be further factorized according to the C-component structure of $\mathcal{G}_{[\mathbf{C}]}$, the subgraph with only variables in $\mathbf{C}$, as stated in the following lemma from [37].

**Lemma 1.** *[C-component decomposition] Let* $\mathbf{C} \subseteq \mathbf{V}$, $\mathbf{C}_1, \ldots, \mathbf{C}_l$ *the C-components of* $\mathcal{G}_{[\mathbf{C}]}$. *Then* $Q[\mathbf{C}] = \prod_j Q[\mathbf{C}_j]$, *and for any topological order* $C_1 < C_2 < C_n$ *of the variables in* $\mathbf{C}$

$$Q[\mathbf{C}_j] = \prod\nolimits_{\{C_i \in \mathbf{C}_j\}} \frac{Q[C_1, \ldots, C_i]}{Q[C_1, \ldots, C_{i-1}]}, \text{ where } Q[C_1, \ldots, C_i] = \sum\nolimits_{c_{i+1}, \ldots, c_n} Q[\mathbf{C}]. \qquad (8)$$

This result allow us to decompose C-factors as products of minimal C-factors. Once both input and output distributions factorized that way, the task is solvable if every factor of the output can mapped to factors in the input distributions. First, we discuss how to match C-factors across different regimes:

**Lemma 2.** *Let* $\mathbf{X}, \mathbf{Z} \subset \mathbf{V}$ *be disjoint sets of variables,* $\sigma_{\mathbf{X}}$ *and* $\sigma_{\mathbf{Z}}$ *be any two interventions, and* $\mathbf{C} \subseteq \mathbf{V}$. *Then,* $Q[\mathbf{C}; \sigma_{\mathbf{X}}, \sigma_{\mathbf{Z}}] = Q[\mathbf{C}; \sigma_{\mathbf{X}}]$ *whenever* $\mathbf{C} \cap \mathbf{Z} = \emptyset$.

For instance, the C-factor $Q^*[Z, Y]$ in Fig. 1(a) is equal to $Q^*[Z, Y; \sigma_{X_1, X_2}]$ because $\{Z, Y\}$ does not intersect $\{X_1, X_2\}$. Similarly, in Fig. 2(d), (e), $Q^1[R, Y; \sigma_Z^1] = Q^1[R, Y; \sigma_Z^1, \sigma_W^2] = Q^1[R, Y]$ because $\{R, Y\}$ is not affected by $\sigma_Z^1$ or $\sigma_W^2$. Next, we consider matching C-factors across domains:

**Lemma 3.** *Let* $\mathcal{G}^{\Delta}$ *be a selection diagram for* $\langle M^k, M^l \rangle$, *then* $Q^k[\mathbf{C}; \sigma_{\mathbf{X}}] = Q^l[\mathbf{C}; \sigma_{\mathbf{X}}]$ *if* $\mathcal{G}^{\Delta}$ *does not contain selection nodes* $S_{v_i}$ *pointing to any variable in* $V_i \in \mathbf{C}$, *that is,* $V_i \notin \Delta^{k,l}$.

From the selection diagram in Fig. 2(c), we can infer $Q^*[X, Z, Y] = Q^1[X, Z, Y] = Q^2[X, Z, Y]$, and $Q^*[W] = Q^2[W]$, but not $Q^*[W] = Q^1[W]$, for example. Next, a query of interest can be written in terms of C-factors based on Thm. 1:

**Corollary 1.** *Let* $\mathbf{Y}, \mathbf{X} \subseteq \mathbf{V}$ *be any two sets of variables, and let* $\sigma_{\mathbf{X}} = \sigma_{\mathbf{X}}^*$ *be an atomic, conditional or stochastic intervention. The effect of* $\sigma_{\mathbf{X}}$ *on* $\mathbf{Y}$ *is given by*

$$P^*(\mathbf{y}; \sigma_{\mathbf{X}} = \sigma_{\mathbf{X}}^*) = \sum\nolimits_{\mathbf{d} \setminus \mathbf{y}} Q^*[\mathbf{X} \cap \mathbf{D}; \sigma_{\mathbf{X}} = \sigma_{\mathbf{X}}^*] Q^*[\mathbf{D} \setminus \mathbf{X}], \qquad (9)$$

*where* $\mathbf{D} = An(\mathbf{Y})_{\mathcal{G}_{\sigma_{\mathbf{X}}}}$. *Furthermore, this effect is transportable from* $\langle \mathcal{G}^{\Delta}, \mathbb{Z} \rangle$ *if and only if* $Q^*[\mathbf{D} \setminus \mathbf{X}]$ *is transportable from* $\langle \mathcal{G}^{\Delta}, \mathbb{Z} \rangle$.

The effect of an intervention conditioned on some evidence $\mathbf{W} = \mathbf{w}$, $P(\mathbf{y}|\mathbf{w}; \sigma_{\mathbf{X}})$, may involve less C-factors than $P(\mathbf{y}, \mathbf{w}; \sigma_{\mathbf{X}})$ depending on the topology of the graph. The exact factorization of the conditional query is given by the following theorem.

**Theorem 2.** *Let* $\mathbf{Y}, \mathbf{X}, \mathbf{W} \subset \mathbf{V}$, $\mathbf{W} \cap \mathbf{Y} = \emptyset$, $\sigma_{\mathbf{X}}$ *be any intervention, and* $\mathcal{G}_{\sigma_{\mathbf{X}}}$ *the corresponding interventional causal graph. Then, the effect of* $\sigma_{\mathbf{X}}$ *on* $\mathbf{Y}$ *conditioned on* $\mathbf{W}$ *is given by*

$$P(\mathbf{y}|\mathbf{w}; \sigma_{\mathbf{X}}) = P(\mathbf{y}|\mathbf{w}_{\mathbf{y}}; \sigma_{\mathbf{X}}, \sigma_{\mathbf{W}_{\overline{\mathbf{y}}}} = \mathbf{w}_{\overline{\mathbf{y}}}) = \sum\nolimits_{\mathbf{a} \setminus (\mathbf{y} \cup \mathbf{w}_{\mathbf{y}})} Q[\mathbf{A}; \sigma_{\mathbf{X}}] \Big/ \sum\nolimits_{\mathbf{a} \setminus \mathbf{w}_{\mathbf{y}}} Q[\mathbf{A}; \sigma_{\mathbf{X}}], \quad (10)$$

*where* $\mathbf{W}_{\mathbf{y}} \subseteq \mathbf{W}$ *is the set of variables in* $\mathbf{W}$ *connected to any* $Y \in \mathbf{Y}$ *by any path (regardless of the directionality) in* $\mathcal{G}_{\sigma_{\mathbf{X}}[\mathbf{D}]_{\underline{\mathbf{W}}}}$, *with* $\mathbf{D} = An(\mathbf{Y} \cup \mathbf{W})_{\mathcal{G}_{\sigma_{\mathbf{X}}}}$, $\mathbf{W}_{\overline{\mathbf{y}}} = \mathbf{W} \setminus \mathbf{W}_{\mathbf{y}}$, *and* $\mathbf{A} = An(\mathbf{Y} \cup \mathbf{W}_{\mathbf{y}})_{\mathcal{G}_{\sigma_{\mathbf{X}} \underline{\mathbf{W}}}}$. *Furthermore, this effect is transportable from* $\langle \mathcal{G}^{\Delta}, \mathbb{Z} \rangle$ *iff* $Q[\mathbf{A}; \sigma_{\mathbf{X}}]$ *is transportable from* $\langle \mathcal{G}^{\Delta}, \mathbb{Z} \rangle$.

**Algorithm 1** $\sigma$-TR$(\mathbf{Y}, \mathbf{W}, \sigma_{\mathbf{X}}, \mathbb{Z}, \mathcal{G}^{\mathbf{\Delta}})$

---

**Input**: $\mathcal{G}^{\mathbf{\Delta}}$ selection diagrams over variables $\mathbf{V}$ for domains $\Pi$; $\mathbf{Y}, \mathbf{W} \subseteq \mathbf{V}$ disjoint subsets of variables; an intervention $\sigma_{\mathbf{X}}^*$ defined over a set $\mathbf{X} \subseteq \mathbf{V}$; and available distribution specification $\mathbb{Z}$.
**Output**: $P^*(\mathbf{y}|\mathbf{w}; \sigma_{\mathbf{X}})$ in terms of available distributions or FAIL if not transportable from $\langle \mathcal{G}^{\mathbf{\Delta}}, \mathbb{Z} \rangle$.

 1: let $\mathbf{A}$ be defined as in Thm. 2, and let $\mathbf{A}_1, \ldots, \mathbf{A}_n$ be the C-components of $\mathcal{G}_{\sigma_{\mathbf{X}}[\mathbf{A}]}$.
 2: **for each** $\mathbf{A}_i$ s.t. $\mathbf{A}_i \cap \mathbf{X} = \emptyset$, $\sigma_{\mathbf{Z}} \in \mathbb{Z}^k \in \mathbb{Z}$ s.t. $\mathbf{A}_i \cap \mathbf{Z} = \emptyset$ and $\mathbf{A}_i \cap \Delta^k = \emptyset$ **do**
 3:     let $\mathbf{B}_i$ be the C-component of $\mathcal{G}_{\sigma_{\mathbf{Z}}}$ such that $\mathbf{A}_i \subseteq \mathbf{B}_i$, compute $Q^k[\mathbf{B}_i; \sigma_{\mathbf{Z}}]$ from $Q^k[\mathbf{V}; \sigma_{\mathbf{Z}}]$.
 4:     **if** IDENTIFY$(\mathbf{A}_i, \mathbf{B}_i, Q[\mathbf{B}_i; \sigma_{\mathbf{Z}}], \mathcal{G}_{\sigma_{\mathbf{Z}}})$ does not FAIL **then**
 5:         let $Q^*[\mathbf{A}_i; \sigma_{\mathbf{X}}] = $ IDENTIFY$(\mathbf{A}_i, \mathbf{B}_i, Q[\mathbf{B}_i; \sigma_{\mathbf{Z}}], \mathcal{G}_{\sigma_{\mathbf{Z}}})$.
 6:         move to the next $\mathbf{A}_i$.
 7:     **end if**
 8: **end for**
 9: **for each** $\mathbf{A}_i$ containing variables in $\mathbf{X}$ let $Q^*[\mathbf{A}_i; \sigma_{\mathbf{X}}] = $ REPLACE$(\mathbf{A}_i, \sigma_{\mathbf{X}})$.
10: **if** any $Q^*[\mathbf{A}_i]$ was not assigned **then return** FAIL.
11: let $Q^*[\mathbf{A}; \sigma_{\mathbf{X}}] = \prod_i Q^*[\mathbf{A}_i; \sigma_{\mathbf{X}}]$.
12: **return** $\sum_{\mathbf{a} \backslash (\mathbf{y} \cup \mathbf{w})} Q^*[\mathbf{A}; \sigma_{\mathbf{X}}] \Big/ \sum_{\mathbf{a} \backslash \mathbf{w}} Q^*[\mathbf{A}; \sigma_{\mathbf{X}}]$.

---

For instance, the effect $P^*(y \mid r, z; \sigma_X = \sigma_X^*)$ in the example of Fig. 2(b) can be transported as

$$P^*(y \mid r, z; \sigma_X) = P^*(y \mid r; \sigma_X, \sigma_Z = z) = Q^*[R, Y; \sigma_X] \Big/ \sum_y Q^*[R, Y; \sigma_X] \,. \tag{11}$$

Using Lemma 3 we have $Q^*[R, Y; \sigma_X] = Q^1[R, Y; \sigma_X]$ and by Lemma 2 equal to $Q^1[R, Y; \sigma_Z]$. From $Q^1[\mathbf{V}; \sigma_Z] = P^1(\mathbf{v}; \sigma_Z)$ we obtain $Q^1[R, Y, W, X; \sigma_Z]$, using Lemma 1, as $Q^1[\mathbf{V}; \sigma_Z]/Q^1[Z; \sigma_Z] = P^1(\mathbf{v}; \sigma_Z)/P^1(z|r, r, x; \sigma_Z) = P^1(y|z, x, r, w; \sigma_Z)P^1(x, r, w; \sigma_Z)$. Then, since $R < Y < W < X$ is a valid topological order in $\mathcal{G}_{[R,Y,W,X]}$, Eq. (8) leads to $Q^1[R, Y; \sigma_Z] = \sum_{x,w} Q^1[R, Y, W, X] = \sum_{x,w} P^1(y|z, x, r, w; \sigma_Z)P^1(x, r, w; \sigma_Z)$. Replacing in Eq. (11) and simplifying results in:

$$P^*(y \mid r, z; \sigma_X) = \sum_{x,w} P^1(y|z, x, r, w; \sigma_Z)P^1(x, w \mid r; \sigma_Z). \tag{12}$$

Building on the observations and results we have so far, we design the algorithm $\sigma$-TR (Alg. 1) that takes as input the variables defining a query, the specification of $\sigma_{\mathbf{X}}$ (i.e., what type of intervention is being applied and its arguments), a set of available distributions ($\mathbb{Z}$), and the selection diagrams. Internally $\sigma$-TR uses the subroutine IDENTIFY from [36] that applies Lemma 1 systematically to obtain a C-factor $Q[\mathbf{A}]$ from $Q[\mathbf{B}]$, where $\mathbf{A} \subseteq \mathbf{B}$, and the subroutine 'REPLACE' to determine the factors of intervened variables according to the particular type of intervention.

Recall the example from the introduction of the paper where the target effect is $P^*(y; \sigma_{X_1, X_2} = \sigma^*)$ corresponding to the regime shown in Fig. 1(d). With $\mathbf{Y} = \{Y\}$, $\mathbf{W} = \emptyset$, $\mathbb{Z} = \{\{\sigma_\emptyset\}, \{\sigma_{X_1, X_2}^1\}, \{\sigma_{X_2}^2\}\}$ and $\mathcal{G}^{\mathbf{\Delta}}$ as in Figs. 1(b), (c); $\sigma$-TR is invoked so that $\mathbf{A} = \{X_1, Z, X_2, Y\}$ with C-components $\mathbf{A}_1 = \{X_1\}$, $\mathbf{A}_2 = \{Z\}$, $\mathbf{A}_3 = \{X_2\}$ and $\mathbf{A}_4 = \{Y\}$ ($\mathcal{G}_{\sigma^*[\mathbf{A}]} = \mathcal{G}_{\sigma^*}$). The loop in line 2 will take $\mathbf{A}_2$ for domain $\pi^k$, $k=1$ and $\sigma_{\mathbf{Z}} = \sigma^1$. $\mathbf{B}_i = \{Z\}$ is the C-component of $\mathcal{G}_{\sigma^1}$ (Fig. 1(b)) that contains $\mathbf{A}_2$, and $Q^1[Z; \sigma^1] = P^1(z \mid x_1; \sigma^1)$ by Lemma 1. Then IDENTIFY is called and returns $Q^*[\mathbf{A}_2] = P^1(z \mid x_1; \sigma^1)$. Similarly, $Q^*[\mathbf{A}_4]$ is obtained from $\pi^2$ with $\mathbf{B}_i = \{Y\}$ as $Q^2[Y; \sigma^2] = P^2(y|x_1, x_2; \sigma^2)$. Next, $Q^*[X_1; \sigma^*]$ and $Q^*[X_2; \sigma^*]$ are given by policy $\sigma^*$ as $P^*(x_1; \sigma^*)$ and $P^*(x_2|z, x_1; \sigma^*)$. Finally, line 12 returns an expression equivalent to Eq. (1).

Determining non-transportability is non-trivial task, however, $\sigma$-TR is both sufficient and necessary:

**Theorem 3.** *[$\sigma$-TR Completeness] The effect $P(\mathbf{y} \mid \mathbf{w}; \sigma_{\mathbf{X}})$ is transportable from $\mathbb{Z}$ in $\mathcal{G}^{\mathbf{\Delta}}$ if and only if the algorithm $\sigma$-TR (Alg. 1) outputs an estimand for it.*

$\sigma$-TR finds an estimand for the query or determines non-transportability efficiently, in $O(n^4 z)$ time, where $n = |\mathbf{V}|$ and $z = \sum_{\pi^i} |\mathbb{Z}^i|$ (see Appendix B.1). Moreover, every step in $\sigma$-TR can be mapped to a derivation with $\sigma$-calculus and probability axioms like the one given in the previous section.

**Corollary 2.** *[$\sigma$-calculus Completeness] The $\sigma$-calculus together with standard probability axioms is complete for the task of transportability with soft interventions.*

## 4 Graphical Characterization

We proved that the procedure $\sigma$-TR, discussed in the previous section, is both sufficient and necessary for transportability task at hand. In this section, we investigate a sufficient and necessary graphical conditions to decide the transportability of a query by graphical inspection.

Consider once more the loan program example in Section 2. If we learn that $R$ also differs from those in $\pi^*$ (i.e., $R$ is also in $\Delta^1$), then the target query is not transportable from the given input distributions. We can examine the reason $\sigma$-TR outputs FAIL for the query $P^*(y; \sigma_X)$. On line 1, since $\mathbf{W}=\emptyset$, $\mathbf{A}=\{R, X, Z\}$ with C-factors $\mathbf{A}_1=\{R, Y\}$, $\mathbf{A}_2=\{Z\}$, and $\mathbf{A}_3=\{X\}$. There are three sources of data in $\mathbb{Z}$, namely, $\mathbb{Z}^*=\{\sigma_\emptyset\}$, $\mathbb{Z}^1=\{\sigma_Z^1\}$ and $\mathbb{Z}^2=\{\sigma_W^2\}$. If $R \in \Delta^1, \Delta^2$, no C-factor including $R$ can be transported from $\mathbb{Z}^1$ or $\mathbb{Z}^2$. Then, the loop in line 2 only runs for $\sigma_\emptyset \in \mathbb{Z}^*$ such that $\mathbf{B}_i=\{W, R, X, Z, Y\}$, and IDENTIFY fails to obtain $Q^*[R, Y]$ from $Q^*[\mathbf{B}_i]$. At line 10, $\sigma$-TR detects that $Q^*[R, Y]$ was not transported and outputs FAIL.

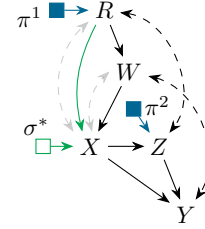

Figure 3: Selection diagram depicting intervention $\sigma_X^*$.

Overall, $\sigma$-TR fails when there exists some C-component $\mathbf{A}_i$ of $\mathcal{G}_{\sigma_\mathbf{X}[\mathbf{A}]}$, a C-component $\mathbf{C}_i$ of $\mathcal{G}^{\boldsymbol{\Delta}}_{[An(\mathbf{A}_i)]}$, and for every $\sigma_\mathbf{Z} \in \mathbb{Z}^k \in \mathbb{Z}$ at least one of the following three conditions occur:

(i) $\mathbf{A}_i \cap \Delta^k \neq \emptyset$, that is, at least one variable in $\mathbf{A}_i$ has a different mechanism in $\pi^k$, or

(ii) $\mathbf{A}_i \cap \mathbf{Z} \neq \emptyset$, meaning at least one of the variables in $\mathbf{A}_i$ have been intervened by $\sigma_\mathbf{Z}$ in $\pi^k$, or

(iii) there exists some $\mathbf{T}_i$ s.t. $\mathbf{A}_i \subset \mathbf{T}_i \subseteq \mathbf{C}_i$, $\mathcal{G}_{\sigma_\mathbf{Z}[\mathbf{T}_i]}$ has a single C-component.

Condition (iii) occurs whenever IDENTIFY (in line 4) fails to obtain $Q^*[\mathbf{A}_i]=Q^k[\mathbf{A}_i; \sigma_\mathbf{Z}]$ from $Q^k[\mathbf{B}_i; \sigma_\mathbf{Z}]$ in $\mathcal{G}_{\sigma_\mathbf{Z}}$ [16, Thm. 3]. The following result follows immediately from Thm. 3:

**Corollary 3.** *Given query $P^*(\mathbf{y}; \sigma_\mathbf{X}=\sigma_\mathbf{X}^*)$, selection diagram $\mathcal{G}^{\boldsymbol{\Delta}}$, and the distribution specified by $\mathbb{Z}$, let $\mathbf{A}$ be defined as in Thm. 2. Then, the query is not transportable from $\langle \mathcal{G}^{\boldsymbol{\Delta}}, \mathbb{Z} \rangle$ if and only if there exists a C-component $\mathbf{A}_i$ of $\mathcal{G}_{\sigma_\mathbf{X}^*[\mathbf{A}]}$ and a C-component $\mathbf{C}_i$ of $\mathcal{G}^{\boldsymbol{\Delta}}_{[An(\mathbf{A}_i)]}$ such that for every $\sigma_\mathbf{Z} \in \mathbb{Z}^k \in \mathbb{Z}$, $\mathbf{A}_i$ satisfies conditions (i), (ii), or (iii).*

For instance consider $\mathcal{G}^{\boldsymbol{\Delta}}$ in Fig. 3 with $\mathbb{Z}=\{\{\sigma_\emptyset\}, \{\sigma_X=x\}, \{\sigma_W=g(R), \sigma_R = P'(R)\}\}$ and intervention $\sigma_X=\sigma_X^*$, where $\sigma_X^*$ enforces some some pre-specified $P^*(X|R, W; \sigma_X^*)$. The edge $R \to X$ is added due to $\sigma_X^*$ while the two bidirected edges in grey are removed by it. Let $\mathbf{A}_i=\{Z, R\}$ and $\mathbf{C}_i=\{R, X, W, Z\}$. We can verify that condition (iii) is satisfied for $\sigma_\emptyset$ from $\pi^*$ and $\sigma_W$ from $\pi^2$. From $\pi^1$, $\sigma_X=x$ cannot be used due to condition (i) and $\sigma_R$ in $\pi^3$ satisfies condition (ii). Although for this instance there is no *s-Thicket* [22] —a graphical structure precluding transportability in the atomic case— for $P(y \mid do(x))$, the effect of $\sigma_X^*$ is not transportable. Instead, the condition in Cor. 3 can be proved equivalent to the presence of an s-Thicket for the effect $P(\mathbf{a} \mid do(\mathbf{x}, an(\mathbf{a}) \setminus \mathbf{a}))$ (see Lemma 10 in Appendix D).

## 5 Conclusions

We studied the problem of transporting effects of soft interventions from knowledge encoded in the form of a selection diagram and a combination of observational and experimental data from multiple, different domains. We showed how the problem can be solved by transporting the effect of an atomic intervention from the same input (Thm. 1). Similarly, we proved that a conditional effect is transportable only if the marginal effect derived through Thm. 2 is transportable. We then developed an efficient algorithm called $\sigma$-TR (Alg. 1) that is both sufficient and necessary for finding a function of the available data that matches the target effect (whenever such a function exists). As a corollary, we conclude that $\sigma$-calculus together with basic probability axioms are complete for the soft transportability task (Cor. 2). Finally, we described a complete graphical condition to determine the transportability of any transportability instance (Cor. 3). We hope that this series of results related to soft interventions, and knowledge of its relationship with atomic interventions, can help data scientists to apply causal inference in broader and more realistic scenarios.

## Broader Impact

Our work investigates the formal conditions under which knowledge acquired in one domain (e.g., setting, population, environment) can be generalized to a different one that may be related, but is unlikely to be the same. This is known in the causal inference literature as the problem of "transportability." As alluded to in the introduction, issues of transportability are pervasive throughout the empirical sciences as well as AI and ML. We believe that having a more foundational tool that allows the empirical investigator to determine whether (and how) her/his understanding of the underlying system is sufficient to support the generalization of an empirical claim is a critical addition to the scientific toolbox. Not having such a tool, on the other hand, may lead researchers to operate on a more heuristical basis, which may lead to a lack of understanding of when things can go wrong and how to fix them. For instance, public policies that will fail or have unintended consequences, potentially harming people, or spending unnecessary societal resources. In the context of automated decision-making in AI, we could have systems following policies that harm the users, can act unfairly, or discriminate against certain groups (e.g., the policy was trained in Scandinavia and moved to the US). By and large, we believe this research on the theory of generalization of policies based on soft interventions can benefit a large group of individuals, including empirical scientists, policy-makers, AI researchers, and society in general.

## Acknowledgments and Disclosure of Funding

This research was supported by grants from IBM Research, Adobe Research, NSF IIS-1704352, and IIS-1750807 (CAREER).

## Footnotes

[1]They acknowledge and discuss the challenges of pursuing such task [12]: "the number of possible variations on a given program is potentially infinite, and a theoretical framework is definitely needed to understand which variations are important to replicate and which are not." Some typical question they try to answer include: "If a program worked for poor rural women in Africa, will it work for middle-income urban men in South Asia?"

[2] The rules of $\sigma$-calculus are provided in Appendix A.

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
