[Supplementary Material]

# General Transportability of Soft Interventions: Completeness Results

## Supplemental Material

## A  Useful Background

### A.1  d-Separation

In Section 2 we use separation statements of the form $(\mathbf{X} \perp\!\!\!\perp \mathbf{Y} \mid \mathbf{Z})$ to validate the application of the rules of $\sigma$-calculus. Those statements can be read of from the graph using the *d-Separation* criterion [14].

**Definition 4** (d-Separation). In a graph $\mathcal{G}$ a path $\overline{p}$ is blocked by a set of nodes $\mathbf{Z}$ if and only if

1. $\overline{p}$ contains a chain of nodes $A \to B \to C$ or a fork $A \leftarrow B \to C$ such that the middle node $B$ is in $\mathbf{Z}$ (i.e., $B$ is conditioned on), or

2. $\overline{p}$ contains a collider $A \to B \leftarrow C$ such that the collision node $B$ is not in $\mathbf{Z}$, and no descendant of $B$ is in $\mathbf{Z}$.

If $\mathbf{Z}$ blocks every path between two nodes $X$ and $Y$, then $X$ and $Y$ are $d$-separated, conditional on $\mathbf{Z}$, that is, $(X \perp\!\!\!\perp Y \mid \mathbf{Z})$.

### A.2  Calculus for Soft Interventions

For the derivation in Sec. 2 we used $\sigma$-calculus [9], a set of rules that allow us to reason about probability expressions with soft interventions. We state it next for reference:

**Theorem 4.** *[Inference Rules – $\sigma$-calculus] Let $\mathcal{G}$ be a causal diagram compatible with a structural causal model $M$, with endogenous variables $\mathbf{V}$. For any disjoint subsets $\mathbf{X}, \mathbf{Y}, \mathbf{Z} \subseteq \mathbf{V}$, two disjoint subsets $\mathbf{T}, \mathbf{W} \subseteq \mathbf{V} \setminus (\mathbf{Z} \cup \mathbf{Y})$ (i.e., possibly including $\mathbf{X}$), the following rules are valid for any intervention strategies $\sigma_{\mathbf{X}}, \sigma_{\mathbf{Z}}$, and $\sigma'_{\mathbf{Z}}$:*

***Rule 1** (Insertion/Deletion of observations):*

$$P(\mathbf{y} \mid \mathbf{w}, \mathbf{t}; \sigma_{\mathbf{X}}) = P(\mathbf{y} \mid \mathbf{w}; \sigma_{\mathbf{X}}) \qquad\qquad \textit{if } (\mathbf{Y} \perp\!\!\!\perp \mathbf{T} \mid \mathbf{W}) \textit{ in } \mathcal{G}_{\sigma_{\mathbf{X}}}. \quad (\text{A.13})$$

***Rule 2** (Change of regimes under observation):*

$$P(\mathbf{y} \mid \mathbf{z}, \mathbf{w}; \sigma_{\mathbf{x}}, \sigma_{\mathbf{z}}) = P(\mathbf{y} \mid \mathbf{z}, \mathbf{w}; \sigma_{\mathbf{x}}, \sigma'_{\mathbf{z}}) \qquad \textit{if } (\mathbf{Y} \perp\!\!\!\perp \mathbf{Z} \mid \mathbf{W}) \textit{ in } \mathcal{G}_{\sigma_{\mathbf{X}} \sigma_{\mathbf{Z}} \underline{\mathbf{Z}}} \textit{ and } \mathcal{G}_{\sigma_{\mathbf{X}} \sigma'_{\mathbf{Z}} \underline{\mathbf{Z}}}. \quad (\text{A.14})$$

***Rule 3** (Change of regimes without observation):*

$$P(\mathbf{y} \mid \mathbf{w}; \sigma_{\mathbf{x}}, \sigma_{\mathbf{z}}) = P(\mathbf{y} \mid \mathbf{w}; \sigma_{\mathbf{x}}, \sigma'_{\mathbf{z}}) \textit{ if } (\mathbf{Y} \perp\!\!\!\perp \mathbf{Z} \mid \mathbf{W}) \textit{ in } \mathcal{G}_{\sigma_{\mathbf{X}} \sigma_{\mathbf{Z}} \overline{\mathbf{Z}(\mathbf{W})}} \textit{ and } \mathcal{G}_{\sigma_{\mathbf{X}} \sigma'_{\mathbf{Z}} \overline{\mathbf{Z}(\mathbf{W})}}, \quad (\text{A.15})$$

*where $\mathbf{Z}(\mathbf{W}) \subseteq \mathbf{Z}$ is the set of elements in $\mathbf{Z}$ that are not ancestors of $\mathbf{W}$ in $\mathcal{G}_{\sigma_{\mathbf{X}}}$.*

## B  Proofs and further discussion for Section 2

### B.1  Detailed derivation of the example in Fig. 2

Here we provide a detail derivation of the causal effect $P^*(y; \sigma_X)$ for the example presented in Section 2. All graphs we refer to are shown in Fig. 4.

First by marginalization:

$$P^*(y; \sigma_X) = \sum_{r,x,z} P^*(y|z, x, r; \sigma_X) P^*(z|x, r; \sigma_X) P^*(x|r; \sigma_X) P^*(r; \sigma_X). \quad (\text{B.16})$$

We will transport factor by factor starting from the last.

By rule 3 and the separation $(R \perp\!\!\!\perp X)$ in $\mathcal{G}_{\sigma_X^* \overline{X}}$ and $\mathcal{G}_{\overline{X}}$, we have

$$P^*(r; \sigma_X) = P^*(r), \tag{B.17}$$

which is estimable from the input distribution $P^*(\mathbf{V})$.

The factor $P^*(x|r; \sigma_X)$ is determined by $\sigma_X^*$ (and the policy's specification).

For the second factor,

$$P^*(z|x, r; \sigma_X) = P^*(z|x, r; \sigma_X, \sigma_W) \tag{B.18}$$

by rule 3 and $(Z \perp\!\!\!\perp W \mid X, R)$ in $\mathcal{G}_{\sigma_X \sigma_W \overline{W}}$ and $\mathcal{G}_{\sigma_X \overline{W}}$. Then, by rule 2 and $(Z \perp\!\!\!\perp X \mid R)$ in $\mathcal{G}_{\sigma_X \sigma_W \underline{X}}$ and $\mathcal{G}_{\sigma_W \underline{X}}$, we obtain

$$P^*(z|x, r; \sigma_X) = P^*(z|x, r; \sigma_W). \tag{B.19}$$

From the graph $\mathcal{G}_{\sigma_W}^{\Delta_2}$, $(Z \perp\!\!\!\perp S_r \mid R, X)$ hence

$$P^*(z|x, r; \sigma_W) = P^2(z|x, r; \sigma_W) \tag{B.20}$$

estimable from the given $P^2(\mathbf{V}; \sigma_W = \sigma_W^2)$.

For the first factor is equal to

$$P^*(y|z, x, r; \sigma_X, \sigma_Z = z) \tag{B.21}$$

by rule 2 and $(Y \perp\!\!\!\perp Z \mid X, R)$ in $\mathcal{G}_{\sigma_X \sigma_Z = z \underline{Z}}$ and $\mathcal{G}_{\sigma_X \underline{Z}}$. We remove the observed $X$ by rule 1 and $(Y \perp\!\!\!\perp X \mid Z, R)$ in $\mathcal{G}_{\sigma_X \sigma_Z = z}$.

$$P^*(y|z, r; \sigma_X, \sigma_Z = z). \tag{B.22}$$

Then, by rule 3 and $(Y \perp\!\!\!\perp X \mid Z, R)$ in $\mathcal{G}_{\sigma_X \sigma_Z = z \overline{X}}$ and $\mathcal{G}_{\sigma_Z = z \overline{X}}$ this is equal to

$$P^*(y|z, r; \sigma_Z = z). \tag{B.23}$$

At this point we can transport this factor from $\pi^1$ due to $(Y \perp\!\!\!\perp S_w \mid Z, R)$ in $\mathcal{G}_{\sigma_Z = z}^{\Delta_1}$:

$$P^1(y|z, r; \sigma_Z). \tag{B.24}$$

We sum over $X$

$$\sum_{x'} P^1(y|z, x', r; \sigma_Z = z) P^1(x' \mid z, r; \sigma_Z = z) \tag{B.25}$$

and use rule 2 with $(Y \perp\!\!\!\perp Z \mid X, R)$ in $\mathcal{G}_{\sigma_Z = z \underline{Z}}$ and $\mathcal{G}_{\sigma_Z z \underline{Z}}$ to exchange $\sigma_Z = z$ with $\sigma_Z = \sigma_Z^1$

$$\sum_{x'} P^1(y|z, x', r; \sigma_Z) P^1(x' \mid z, r; \sigma_Z = z). \tag{B.26}$$

Due to $(X \perp\!\!\!\perp Z)$ in $\mathcal{G}_{\sigma_Z = z}$ the observation of $Z$ can be removed in the second factor in the sum:

$$\sum_{x'} P^1(y|z, x', r; \sigma_Z) P^1(x' \mid r; \sigma_Z = z). \tag{B.27}$$

Note that $(X \perp\!\!\!\perp Z)$ in $\mathcal{G}_{\sigma_Z = z \overline{Z}}$ and $\mathcal{G}_{\sigma_Z \overline{Z}}$, therefore by rule 3 we get

$$\sum_{x'} P^1(y|z, x', r; \sigma_Z) P^1(x' \mid r; \sigma_Z), \tag{B.28}$$

resulting in a sum estimable from $P^1(\mathbf{V}; \sigma_Z)$.

Putting all together, we have:

$$P^*(y; \sigma_X) = \sum_{r,x,z} \underbrace{\left( \sum_{x'} P^1(y|z, x', r; \sigma_Z) P^1(x'|r; \sigma_Z) \right)}_{\text{from } \sigma_Z^1 \text{ in } \pi^1} \underbrace{P^2(z|x, r; \sigma_W)}_{\text{from } \sigma_W^2 \text{ in } \pi^2} \underbrace{P^*(x|r; \sigma_X^*)}_{\text{def. } \sigma_X^*} \underbrace{P^*(r)}_{\text{from } \pi^*}. \tag{B.29}$$

Figure 4: Graphs used in the derivation of the example in Sec. 2. Edges added by intervention are shown in green, those removed by intervention in grey, and those cut due to the application of the rule are shown in faded red.

## B.2  Proof of the Reduction

**Theorem 1.** *Let $\mathbf{Y}, \mathbf{X} \subseteq \mathbf{V}$ be any two sets of variables, and let $\sigma_{\mathbf{X}}^*$ be an atomic, conditional or stochastic intervention. Then, the effect of $\sigma_{\mathbf{X}}^*$ on $\mathbf{Y}$ can be written as*

$$P^*(\mathbf{y}; \sigma_{\mathbf{X}}=\sigma_{\mathbf{X}}^*) = \sum_{\mathbf{d}\backslash\mathbf{y}} P^*(\mathbf{d} \backslash \mathbf{x}; \sigma_{\mathbf{X}}=\mathbf{x}) \prod_{X\in\mathbf{X}\cap\mathbf{D}} P^*(x \mid \mathbf{pa}_x; \sigma_{\mathbf{X}}=\sigma_{\mathbf{X}}^*). \qquad (5)$$

*Moreover, this effect is transportable from $\langle \mathcal{G}^{\boldsymbol{\Delta}}, \mathbb{Z} \rangle$ if and only if $P^*(\mathbf{d} \backslash \mathbf{x}; \sigma_{\mathbf{X}}=\mathbf{x})$ is transportable from $\langle \mathcal{G}^{\boldsymbol{\Delta}}, \mathbb{Z} \rangle$, where $\mathbf{D} = An(\mathbf{Y})_{\mathcal{G}_{\sigma_{\mathbf{X}}}}$.*

*Proof.* We can start by summing over the $\mathbf{D} \backslash \mathbf{Y}$:

$$P^*(\mathbf{y}; \sigma_{\mathbf{X}}=\sigma_{\mathbf{X}}^*) = \sum_{\mathbf{d}\backslash\mathbf{y}} P^*(\mathbf{d}; \sigma_{\mathbf{X}}=\sigma_{\mathbf{X}}^*). \qquad (\text{B.30})$$

Let $V_1 < V_2 < \ldots$ be a topological order of the variables in $\mathbf{D}$ in the graph $\mathcal{G}_{\sigma_{\mathbf{X}}}$ and let $\mathbf{V}^{<i}$ be the set of variables in $\mathbf{D}$ that comes before $V_i$ in the order. Then we can write

$$P^*(\mathbf{y}; \sigma_{\mathbf{X}}=\sigma_{\mathbf{X}}^*) = \sum_{\mathbf{d}\backslash\mathbf{y}} \prod_{V_i\in\mathbf{D}} P^*(v_i \mid \mathbf{v}^{<i}; \sigma_{\mathbf{X}}=\sigma_{\mathbf{X}}^*). \qquad (\text{B.31})$$

For every $V_i \notin \mathbf{X}$ let $\mathbf{X}^{<i} = \mathbf{X} \cap \mathbf{V}^{<i}$ and $\mathbf{X}^{>i} = \mathbf{X} \backslash \mathbf{V}^{<i}$. First we will use rule 3 of $\sigma$-calculus to exchange $\sigma_{\mathbf{X}}^*$ with an atomic intervention for the variables in $\mathbf{X}$ that go after $V_i$. This is allowed by $(V_i \perp\!\!\!\perp \mathbf{X}^{>i} \mid \mathbf{V}^{<i})$ in both $\mathcal{G}_{\sigma_{\mathbf{X}}\overline{\mathbf{X}^{>i}}}$ and $\mathcal{G}_{\sigma_{\mathbf{X}^{<i}}(\sigma_{\mathbf{X}^{>i}}=\mathbf{x}^{>i})\overline{\mathbf{X}^{>i}}}$.

$$P^*(v_i \mid \mathbf{v}^{<i}; \sigma_{\mathbf{X}}=\sigma_{\mathbf{X}}^*) = P^*(v_i \mid \mathbf{v}^{<i}; \sigma_{\mathbf{X}^{<i}}=\sigma_{\mathbf{X}^{<i}}^*, \sigma_{\mathbf{X}^{>i}}=\mathbf{x}^{>i}), \qquad (\text{B.32})$$

In other words, $V_i$ is not affected by interventions on variables that come before as long as we don't condition on their descendants.

Next, we will do the same for variables in $\mathbf{X}^{>i}$. The difference is that those are observed because they belong to $\mathbf{V}^{<i}$. The statement $(V_i \perp\!\!\!\perp \mathbf{X}^{<i} \mid \mathbf{V}^{<i} \backslash \mathbf{X})$ holds in $\mathcal{G}_{(\sigma_{\mathbf{X}^{>i}}=\mathbf{x}^{>i})\sigma_{\mathbf{X}^{<i}}\underline{\mathbf{X}^{<i}}}$ and $\mathcal{G}_{(\sigma_{\mathbf{X}^{>i}}=\mathbf{x}^{>i})(\sigma_{\mathbf{X}^{<i}}=\mathbf{x}^{<i})\underline{\mathbf{X}^{<i}}}$. To see why consider any $X \in \mathbf{X}^{<i}$ that may be connected to $V_i$ by an active path in those graphs. The path must have arrows into $X$ and the arrow is not bidirected because under intervention $\sigma_X$ no such edge appears in the graph. Then, the arrow must be direct and the observable variable at the tail is in $\mathbf{V}^{<i}$, so the path is blocked. By those separations and rule 2 it follows:

$$P^*(v_i \mid \mathbf{v}^{<i}; \sigma_{\mathbf{X}}=\sigma_{\mathbf{X}}^*) = P^*(v_i \mid \mathbf{v}^{<i}; \sigma_{\mathbf{X}^{<i}}=\mathbf{x}^{<i}, \sigma_{\mathbf{X}^{>i}}=\mathbf{x}^{>i}) \qquad (\text{B.33})$$

$$= P^*(v_i \mid \mathbf{v}^{<i}; \sigma_{\mathbf{X}}=\mathbf{x}). \qquad (\text{B.34})$$

At this point all interventions are atomic. Using the definition of conditional probability:

$$P^*(v_i \mid \mathbf{v}^{<i}; \sigma_{\mathbf{X}}=\mathbf{x}) = \frac{P^*(v_i \mid \mathbf{v}^{<i} \backslash \mathbf{x}^{<i}; \sigma_{\mathbf{X}}=\mathbf{x})P^*(\mathbf{x}^{<i} \mid v_i, \mathbf{v}^{<i} \backslash \mathbf{x}^{<i}; \sigma_{\mathbf{X}}=\mathbf{x})}{P^*(\mathbf{x}^{<i} \mid \mathbf{v}^{<i} \backslash \mathbf{x}^{<i}; \sigma_{\mathbf{X}}=\mathbf{x}).} \qquad (\text{B.35})$$

The second factor in the numerator and the denominator are equal to 1 because under $\sigma_{\mathbf{X}}=\mathbf{x}$ the intervened variables always take the specified values, in summary

$$P^*(v_i \mid \mathbf{v}^{<i}; \sigma_{\mathbf{X}}=\sigma_{\mathbf{X}}^*) = P^*(v_i \mid \mathbf{v}^{<i} \backslash \mathbf{x}^{<i}; \sigma_{\mathbf{X}}=\mathbf{x}). \qquad (\text{B.36})$$

Then we can write the original effect as

$$P^*(\mathbf{y}; \sigma_{\mathbf{X}}=\sigma_{\mathbf{X}}^*) = \sum_{\mathbf{d}\backslash\mathbf{y}} \prod_{V_i\in\mathbf{D}\backslash\mathbf{X}} P^*(v_i \mid \mathbf{v}^{<i} \backslash \mathbf{x}^{<i}; \sigma_{\mathbf{X}}=\mathbf{x}) \prod_{V_i\in\mathbf{D}\cap\mathbf{X}} P^*(v_i \mid \mathbf{v}^{<i}; \sigma_{\mathbf{X}}=\sigma_{\mathbf{X}}^*).$$
$$(\text{B.37})$$

The first product can be simply written as $P^*(\mathbf{d} \backslash \mathbf{x}; \sigma_{\mathbf{X}}=\mathbf{x})$ by virtue of the product/chain rule of probabilities. In the second product, for every $V_i \in \mathbf{X}$ we have $(V_i \perp\!\!\!\perp \mathbf{V}^{<i} \backslash \mathbf{Pa}_i \mid \mathbf{Pa}_i)$. Finally we get

$$P^*(\mathbf{y}; \sigma_{\mathbf{X}}=\sigma_{\mathbf{X}}^*) = \sum_{\mathbf{d}\backslash\mathbf{y}} P^*(\mathbf{d} \backslash \mathbf{x}; \sigma_{\mathbf{X}}=\mathbf{x}) \prod_{X\in\mathbf{D}\cap\mathbf{X}} P^*(x \mid \mathbf{pa}_x; \sigma_{\mathbf{X}}=\sigma_{\mathbf{X}}^*), \qquad (\text{B.38})$$

which corresponds to Eq. (5).

Since every variable in $\mathbf{X}$ has its own C-component in $\mathcal{G}_{\sigma_\mathbf{X}}$, Lemma 1 give us

$$Q^*[\mathbf{D}; \sigma_\mathbf{X}^*] = Q^*[\mathbf{D} \setminus \mathbf{X}; \sigma_\mathbf{X}^*] Q^*[\mathbf{X} \cap \mathbf{D}; \sigma_\mathbf{X}^*]. \tag{B.39}$$

Then by Lemma 2,

$$Q^*[\mathbf{D} \setminus \mathbf{X}; \sigma_\mathbf{X}^*] = Q^*[\mathbf{D} \setminus \mathbf{X}] = Q^*[\mathbf{D} \setminus \mathbf{X}; \sigma_\mathbf{X}=\mathbf{x}]. \tag{B.40}$$

If $Q^*[\mathbf{D} \setminus \mathbf{X}; \sigma_\mathbf{X}^*]$ is not transportable $Q^*[\mathbf{D}; \sigma_\mathbf{X}=\mathbf{x}]$ is also not transportable, because whenever the latter is transportable the former can be computed via Eq.(B.39). Moreover, $Q^*[\mathbf{D} \setminus \mathbf{X}; \sigma_\mathbf{X}=\mathbf{x}]$ is the same as $P^*(\mathbf{d} \setminus \mathbf{x}; \sigma_\mathbf{X}=\mathbf{x})$.

Then, to show the necessity $P^*(\mathbf{d} \setminus \mathbf{x}; \sigma_\mathbf{X}=\mathbf{x})$ consider Thm. 2 with $\mathbf{W} = \emptyset$. In this case this case $\mathbf{A} = An(\mathbf{Y})_{\mathcal{G}_{\sigma_\mathbf{X}}} = \mathbf{D}$, then we have that the effect of $P^*(\mathbf{y}; \sigma_\mathbf{X}=\sigma_\mathbf{X}^*)$ is transportable if and only if $Q^*[\mathbf{D}; \sigma_\mathbf{X}]$ is transportable. $\qquad\square$

## B.3 Subtleties of using the reduction with do-calculus or transportability algorithms for atomic interventions

As discussed towards the end of Sec. 2 known algorithms for transportability are not well-suited to produce estimands as functions of soft-experimental interventions.

Consider GTR in [23] which is the closest algorithm to ours. Clearly, it cannot be used directly because there is mismatch in the format of the inputs due to the non-atomic nature of the interventions in the query and available distributions. Queries such as $P^*(\mathbf{y}, \mathbf{x}; \sigma_\mathbf{X})$ are usually trivial in the case of atomic interventions where they are either $0$ or $1$ depending on whether $\mathbf{x}$ is consistent with the constants enforced by the atomic intervention, yet they could have meaning in a soft-intervention setting.

For the query side the possible issues can be solved via the reduction that we have provided in Thm. 1 which offers a mapping between a marginal soft-intervention query to a marginal atomic-interventions query. However, conditional queries need extra care. For instance, $P^*(\mathbf{y}; \sigma_\mathbf{X})$ is a different query that $P^*(\mathbf{y} \mid \mathbf{x}; \sigma_\mathbf{X})$. Actually, they are guaranteed to match only for *do*-interventions, as shown next:

$$P(\mathbf{y} \mid \mathbf{x}; \sigma_\mathbf{X}=\mathbf{x}) = \frac{P(\mathbf{y}, \mathbf{x}; \sigma_\mathbf{X}=\mathbf{x})}{P(\mathbf{x}; \sigma_\mathbf{X}=\mathbf{x})} \tag{B.41}$$

$$= \frac{P(\mathbf{y}; \sigma_\mathbf{X}=\mathbf{x})P(\mathbf{x} \mid \mathbf{y}; \sigma_\mathbf{X}=\mathbf{x})}{P(\mathbf{x}; \sigma_\mathbf{X}=\mathbf{x})} \tag{B.42}$$

$$= \frac{P(\mathbf{y}; \sigma_\mathbf{X}=\mathbf{x})(1)}{(1)} \tag{B.43}$$

$$= P(\mathbf{y}; \sigma_\mathbf{X}=\mathbf{x}). \tag{B.44}$$

This equality does not hold in general for any intervention, and particularly it does not hold for $\sigma_\mathbf{X}=\emptyset$. Similarly, other variations such as $P^*(\mathbf{y} \mid \mathbf{w}; \sigma_\mathbf{X})$ and $P^*(\mathbf{y} \mid \mathbf{w}, \mathbf{x}; \sigma_\mathbf{X})$ are also not the same. Note that in $\sigma$-TR the set $\mathbf{W}$ could intersect $\mathbf{X}$. In this case Thm. 2 could be used to bypass GTR and use its marginal version directly GTRU "truncating" the available experiments by pretending they are from atomic interventions.

Nevertheless, when GTR outputs terms with "truncated" experimental distributions, care is needed to translate those expressions into their original soft-intervention counterparts. In Sec. 2 we described how for the instance in Fig. 1 GTRU produces the following result when used after the reductions described so far:

$$P^*(y; \sigma_{X_1,X_2}^*) = \sum_{x_1,z,x_2} P^2(y|x_1, do(x_2))P^1(z|do(x_1,x_2))P(x_2|z, x_1; \sigma^*)P(x_1; \sigma^*). \tag{B.45}$$

As we saw from Eq. (1) the second factor should be $P^2(z \mid x_1; \sigma_{X_1,X_2}=\sigma_{X_1,X_2}^1)$. Notice that this term is not necessarily equal to $P^1(z; \sigma^1)$ nor $P^1(z|x_1, x_2; \sigma^1)$, so even though the intervention is on both $X_1$ and $X_2$, only $X_1$ should appear as an observation after the translation is done. The same reasoning needs to be applied to every term.

The topics of discussion in this section motivated us to propose $\sigma$-TR as a straightforward algorithm that can handle the task from end to end and subsumes GTR and previous algorithms such as GID [22], Z-ID [3] and ID, IDC [32, 33].

## C  Proofs for Section 3

**Lemma 2.** *Let* $\mathbf{X}, \mathbf{Z} \subset \mathbf{V}$ *be disjoint sets of variables,* $\sigma_{\mathbf{X}}$ *and* $\sigma_{\mathbf{Z}}$ *be any two interventions, and* $\mathbf{C} \subseteq \mathbf{V}$. *Then,* $Q[\mathbf{C}; \sigma_{\mathbf{X}}, \sigma_{\mathbf{Z}}] = Q[\mathbf{C}; \sigma_{\mathbf{X}}]$ *whenever* $\mathbf{C} \cap \mathbf{Z} = \emptyset$.

*Proof.* (Adapted from [9]) By definition of $Q[\mathbf{C}; \sigma_{\mathbf{X}}, \sigma_{\mathbf{Z}}]$ (Eq. (7)) we have

$$Q[\mathbf{C}; \sigma_{\mathbf{X}}, \sigma_{\mathbf{Z}}](\mathbf{v}) = \sum\nolimits_{\mathbf{u}(\mathbf{C})} \prod\nolimits_{\{i | V_i \in \mathbf{C}\}} P(v_i \mid \mathbf{pa}_i, \mathbf{u}_i; \sigma_{\mathbf{X}}, \sigma_{\mathbf{Z}}) P(u(\mathbf{C}); \sigma_{\mathbf{X}}, \sigma_{\mathbf{Z}}), \quad \text{(C.46)}$$

since no variable in $\mathbf{C}$ is in $\mathbf{Z}$ every term $P(v_i \mid \mathbf{pa}_i, \mathbf{u}_i; \sigma_{\mathbf{X}}, \sigma_{\mathbf{Z}}) = P(v_i \mid \mathbf{pa}_i, \mathbf{u}_i; \sigma_{\mathbf{X}})$ and interventions cannot affect the distribution of variables in $\mathbf{U}$, hence $P(\mathbf{u}(\mathbf{C}); \sigma_{\mathbf{X}}, \sigma_{\mathbf{Z}}) = P(\mathbf{u}(\mathbf{C}); \sigma_{\mathbf{X}}) = P(\mathbf{u}(\mathbf{C}))$. Replacing those terms, we obtain exactly the definition of $Q[\mathbf{C}](\mathbf{v})$. $\square$

**Lemma 3.** *Let* $\mathcal{G}^{\Delta}$ *be a selection diagram for* $\langle M^k, M^l \rangle$, *then* $Q^k[\mathbf{C}; \sigma_{\mathbf{X}}] = Q^l[\mathbf{C}; \sigma_{\mathbf{X}}]$ *if* $\mathcal{G}^{\Delta}$ *does not contain selection nodes* $S_{v_i}$ *pointing to any variable in* $V_i \in \mathbf{C}$, *that is,* $V_i \notin \Delta^{k,l}$.

*Proof.* From the definition $Q[.]$ we have

$$Q^{(j)}[\mathbf{C}; \sigma_{\mathbf{X}}](\mathbf{v}) = \sum_{\mathbf{u}(\mathbf{C})} \prod_{\{i | V_i \in \mathbf{C}\}} P^{(j)}(v_i \mid \mathbf{pa}_i, \mathbf{u}_i; \sigma_{\mathbf{X}}) P^{(j)}(\mathbf{u}(\mathbf{C})), \quad j = k, l. \quad \text{(C.47)}$$

A selection node $S_i$ in $\mathcal{G}^{\Delta}$ points to a variable in $V_i \in \mathbf{C}$ only if $f_i^{(k)} \neq f_i^{(l)}$ or if $P^{(k)}(\mathbf{U}_i) \neq P^{(l)}(\mathbf{U}_i)$. Thus, the absence of $S_i \rightarrow V_i$ implies $P^{(k)}(v_i \mid \mathbf{pa}_i, \mathbf{u}_i; \sigma_{\mathbf{X}}) = P^{(l)}(v_i \mid \mathbf{pa}_i, \mathbf{u}_i; \sigma_{\mathbf{X}})$ and $P^{(k)}(\mathbf{U}_i) = P^{(l)}(\mathbf{U}_i)$. Then, every term in equation (C.47) is the same in both domains and the claim follows. $\square$

**Corollary 1.** *Let* $\mathbf{Y}, \mathbf{X} \subseteq \mathbf{V}$ *be any two sets of variables, and let* $\sigma_{\mathbf{X}} = \sigma_{\mathbf{X}}^*$ *be an atomic, conditional or stochastic intervention. The effect of* $\sigma_{\mathbf{X}}$ *on* $\mathbf{Y}$ *is given by*

$$P^*(\mathbf{y}; \sigma_{\mathbf{X}} = \sigma_{\mathbf{X}}^*) = \sum\nolimits_{\mathbf{d} \setminus \mathbf{y}} Q^*[\mathbf{X} \cap \mathbf{D}; \sigma_{\mathbf{X}} = \sigma_{\mathbf{X}}^*] Q^*[\mathbf{D} \setminus \mathbf{X}], \quad \text{(9)}$$

*where* $\mathbf{D} = An(\mathbf{Y})_{\mathcal{G}_{\sigma_{\mathbf{X}}}}$. *Furthermore, this effect is transportable from* $\langle \mathcal{G}^{\Delta}, \mathbb{Z} \rangle$ *if and only if* $Q^*[\mathbf{D} \setminus \mathbf{X}]$ *is transportable from* $\langle \mathcal{G}^{\Delta}, \mathbb{Z} \rangle$.

*Proof.* From [38] we have that $Q^*[\mathbf{D} \setminus \mathbf{X}] = P^*(\mathbf{d} \setminus \mathbf{x}; \sigma_{\mathbf{X}} = \mathbf{x})$ and by definition of C-factor:

$$Q^*[\mathbf{X}; \sigma_{\mathbf{X}} = \sigma_{\mathbf{X}}^*] = \sum_{\mathbf{u_x}} \prod_{X \in \mathbf{X}} P(x \mid \mathbf{pa}_x, \mathbf{u}_x; \sigma_{\mathbf{X}} = \sigma_{\mathbf{X}}^*) P(\mathbf{u_x}). \quad \text{(C.48)}$$

Since the unobservable parents for intervened variables are assumed to be disjoint from any other such set,

$$Q^*[\mathbf{X}; \sigma_{\mathbf{X}} = \sigma_{\mathbf{X}}^*] = \prod_{X \in \mathbf{X}} \sum_{\mathbf{u}_x} P^*(x \mid \mathbf{pa}_x, \mathbf{u}_x; \sigma_{\mathbf{X}} = \sigma_{\mathbf{X}}^*) P^*(\mathbf{u}_x) \quad \text{(C.49)}$$

$$= \prod_{X \in \mathbf{X}} P^*(x \mid \mathbf{pa}_x; \sigma_{\mathbf{X}} = \sigma_{\mathbf{X}}^*). \quad \text{(C.50)}$$

This proves that Eq. (9) is equal to Eq. (5) and implies the claim. $\square$

### C.1  Conditional Queries

#### C.1.1  Auxiliary lemmas

**Lemma 4.** *Suppose* $Q[\mathbf{A}; \sigma_{\mathbf{X}}]$ *is not identifiable from a set of available distributions in a causal diagram* $\mathcal{G}$. *Let* $A_1, A_2 \in \mathbf{A}$ *such that there exists an edge* $A_1 \rightarrow A_2$ *in* $\mathcal{G}$. *Then* $\sum_{a_1} Q[\mathbf{A}]$ *is not identifiable from the same input either.*

*Proof.* Let $\mathcal{M}_1$ and $\mathcal{M}_2$ be the two models witnessing the non-identifiability of $Q[\mathbf{A}; \sigma_{\mathbf{X}}]$, they agree on available distributions, but for some value-assignment $\mathbf{v}'$ we have $Q^1[\mathbf{A}; \sigma_{\mathbf{X}}](\mathbf{v}') = \alpha$, $Q^2[\mathbf{A}; \sigma_{\mathbf{X}}](\mathbf{v}') = \beta$ with $\alpha \neq \beta$. Assume, without loss of generality that $\alpha > \beta$. We will extend a strategy used by [17] to construct two models $\mathcal{M}_1'$ and $\mathcal{M}_2'$ where the domain of $A_2$ is $\mathcal{D}_{A_2} \times \{0, 1\}$, where $\mathcal{D}_{A_2}$ is the domain of $A_2$ in $\mathcal{M}_1, \mathcal{M}_2$. Let $F(A_1)$ be a probability function from $\mathcal{D}_{A_1}$ to $\{0, 1\}$, such that $P(F(a_1) = i) > 0, i = 0, 1$ and $P(F(a_1) = 0) = 1 - P(F(a_1) = 1)$. In $\mathcal{M}_i', i = 1, 2$ we define:

$$P_i^{M_i'}((a_2, k) \mid \mathbf{pa}_{a_2}, u_{a_2}) = P^{M_i}(a_2 \mid \mathbf{pa}_{a_2}, u_{a_2})P(F(a_1){=}k). \tag{C.51}$$

And for $V_j \in \mathbf{V} \setminus \{A_2\}$ let $P^{M_i'}(v_j|\mathbf{pa}_j, u_j) = P^{M_i}(v_j|\mathbf{pa}_j, u_j)$. We can verify that for any $\sigma_{\mathbf{Z}_j} \in \mathbb{Z}$

$$P^{M_1'}(\mathbf{v} \setminus a_2, (a_2, k); \sigma_{\mathbf{Z}_j})$$
$$= Q^{M_1'}[\mathbf{V} \setminus \{A_2\}, (A_2, K); \sigma_{\mathbf{Z}_j}](\mathbf{v} \setminus a_2, (a_2, k)) \tag{C.52}$$
$$= Q^{M_1}[\mathbf{V} \setminus \{A_2\}, (A_2, K); \sigma_{\mathbf{Z}_j}](\mathbf{v})P(F(a_1) = k) \tag{C.53}$$
$$= Q^{M_2}[\mathbf{V} \setminus \{A_2\}, (A_2, K); \sigma_{\mathbf{Z}_j}](\mathbf{v})P(F(a_1) = k) \tag{C.54}$$
$$= Q^{M_2'}[\mathbf{V} \setminus \{A_2\}, (A_2, K); \sigma_{\mathbf{Z}_j}](\mathbf{v} \setminus a_2, (a_2, k)) \tag{C.55}$$
$$= P^{M_2'}(\mathbf{v} \setminus a_2, (a_2, k); \sigma_{\mathbf{Z}_j}). \tag{C.56}$$

Consider the assignment $\mathbf{v}' \setminus \{a_2\}, (a_2', 0)$, by construction we have

$$Q^{M_i'}[\mathbf{A}; \sigma_{\mathbf{X}}](\mathbf{v}' \setminus \{a_2\}, (a_2', 0))$$
$$= Q^{M_i}[\mathbf{A}; \sigma_{\mathbf{X}}](\mathbf{v}')P(F(a_1') = 0). \tag{C.57}$$

Let $P(F(a_1') = 0) = 1/2$ and $P(F(a_1) = 0) = (\alpha - \beta)/4$, for $a_1 \neq a_1'$. It yields:

$$\sum_{a_1} Q^{M_i'}[\mathbf{A}; \sigma_{\mathbf{X}}](\mathbf{v}' \setminus \{a_1, a_2\}, (a_2', 0))$$
$$= \sum_{a_1} Q^{M_i}[\mathbf{A}; \sigma_{\mathbf{X}}](\mathbf{v}' \setminus \{a_1, a_2\})P(F(a_1) = 0) \tag{C.58}$$

For $\mathcal{M}_1'$ this means

$$\sum_{a_1} Q^{M_1'}[\mathbf{A}; \sigma_{\mathbf{X}}](\mathbf{v}' \setminus \{a_1, a_2\}, (a_2', 0))$$
$$= \tfrac{1}{2}\alpha + \left(\tfrac{\alpha - \beta}{4}\right) \sum_{a_1 \neq a_1'} Q^{M_i}[\mathbf{A}; \sigma_{\mathbf{X}}](\mathbf{v}' \setminus \{a_1, a_2\}) \tag{C.59}$$
$$> \tfrac{1}{2}\alpha \tag{C.60}$$

As for $\mathcal{M}_2'$:

$$\sum_{a_1} Q^{M_1'}[\mathbf{A}; \sigma_{\mathbf{X}}](\mathbf{v}' \setminus \{a_1, a_2\}, (a_2', 0))$$
$$= \tfrac{1}{2}b + \left(\tfrac{a - b}{4}\right) \sum_{a_1 \neq a_1'} Q^{M_i}[\mathbf{A}; \sigma_{\mathbf{X}}](\mathbf{v}' \setminus \{a_1, a_2\}) \tag{C.61}$$
$$< \tfrac{1}{2}\beta + \tfrac{\alpha - \beta}{4} \tag{C.62}$$
$$< \tfrac{1}{2}\alpha. \tag{C.63}$$

Then, $\mathcal{M}_1'$ and $\mathcal{M}_2'$ are compatible with $\mathcal{G}$, match in the available distributions and yield different $\sum_{a_1} Q[\mathbf{A}; \sigma_{\mathbf{X}}]$. $\qquad\square$

**Lemma 5.** *Let $A, B$ and $C$ be binary random variables causally related as given by the chain $A \to B \to C$. And suppose $P(B = 1 \mid A = 1) = \alpha$ and $P(B = 1 \mid A = 0) = 1 - \alpha$, for some $\alpha \in [0, 1]$. Then, for any $\beta$ such that $|1/2 - \beta| \leq |1/2 - \alpha|$ there is always a function $f_C$ such that $P(C = 1 \mid A = 1) = \beta$, $P(C = 1 \mid A = 0) = 1 - \beta$ and $P(A, B, C)$ is a positive distribution.*

*Proof.* Let $P(C = 1 \mid B = 1) = x$ and $P(C = 1 \mid B = 0) = 1 - x$, then

$$\beta = P(C = 1 \mid A = 1) = \sum_b P(C = 1 \mid b)P(b \mid A = 1) = 1 - \alpha + x(2\alpha - 1) \qquad \text{(C.64)}$$

$$x = \frac{\alpha + \beta - 1}{2\alpha - 1}. \qquad \text{(C.65)}$$

Since $x$ must belong to the interval $(0, 1)$, we can bound $\beta \in (1 - \alpha, \alpha)$ if $\alpha \leq 1/2$ and $\beta \in (\alpha, 1 - \alpha)$ if $\alpha > 1/2$. Both conditions are satisfied when $|1/2 - \beta| \leq |1/2 - \alpha|$ as assumed, so any solution $x$ is a valid probability.

Then, we can define the function for $C$ as

$$f_C = B \oplus U_c, \qquad \text{(C.66)}$$

where, $U_c$ is a binary unobservable, $P(U_c = 0) = x$ and $\oplus$ is the logical xor operator. $\qquad \square$

**Lemma 6.** *Let $A$ and $B$ be two variables in a causal graph where $A \leftarrow A_n \leftarrow A_{n-1} \leftarrow \cdots \leftarrow A_1 \leftarrow C \rightarrow B_1 \rightarrow \cdots \rightarrow B_{m-1} \rightarrow B_m \rightarrow B$. The variables $A_1, \ldots, A_n, B_1, \ldots, B_m$ are observable, $C$ could be observable or unobservable and $m, n$ are non-negative integers. Then we can define functions for all variables involved such that they are binary and*

$$P(a, b) = \begin{cases} \frac{1}{2}\gamma & \text{if } a = b \\ \frac{1}{2}(1 - \gamma) & \text{otherwise} \end{cases}, \qquad \text{(C.67)}$$

*for any $\gamma \in (0, 1)$.*

*Proof.* First, if $C$ is unobservable set $P(C = 1) = 1/2$, else define an unobservable $U_c$ with $P(U_c = 1) = 1/2$ and let $f_C = U_c$. Let $\alpha, \beta \in (0, 1)$ be parameters to decide later.

If $n = 0$ define $f_A$ such that $P(A = 1 \mid C = 1) = \alpha$, $P(A = 1 \mid C = 0) = 1 - \alpha$. Similarly, if $m = 0$ define $f_B$ such that $P(B = 1 \mid C = 1) = \alpha$, $P(B = 1 \mid C = 0) = 1 - \alpha$.

Suppose $n > 0$, then we will define the functions for $A_1, A_2, \ldots, A_n, A$ such that $P(A_i = 1 \mid C = 1)$ gets closer to $\alpha$ as $i$ increases. If $\alpha < 1/2$, set $f_{A_1}$ such that $P(A_1 = 1 \mid C = 1) = \alpha/(n + 1)$, $P(A_1 = 1 \mid C = 0) = 1 - \alpha/(n + 1)$. Then use lemma 5 to define $f_{A_i}$, $i = 1, \ldots, n$ and $f_A$ such that $P(A_i = 1 \mid C = 1) = i\alpha/(n + 1)$, $P(A_i = 1 \mid C = 0) = 1 - i\alpha/(n + 1)$ and finally $P(A = 1 \mid C = 1) = (n + 1)\alpha/(n + 1) = \alpha$, $P(A = 1 \mid C = 1) = 1 - \alpha$.

If $\alpha > 1/2$ use the same strategy but starting from $P(A_1 = 1 \mid C = 1) = 1 - (1 - \alpha)/(n + 1)$ and decreasing as $P(A_i = 1 \mid C = 1) = 1 - i(1 - \alpha)/(n + 1)$, to obtain $P(A = 1 \mid C = 1) = 1 - (n + 1)(1 - \alpha)/(n + 1) = \alpha$.

The same procedure is applied for $B_1, \ldots, B_m, B$ to obtain $P(B = 1 \mid C = 1) = \beta$, $P(B = 1 \mid C = 0) = 1 - \beta$.

Finally,

$$P(A = 1, B = 1) = \sum_c P(A = 1 \mid c)P(B = 1 \mid c)P(c) \qquad \text{(C.68)}$$

$$= \tfrac{1}{2}[\alpha\beta + (1 - \alpha)(1 - \beta)] \qquad \text{(C.69)}$$

$$P(A = 0, B = 0) = \tfrac{1}{2}[(1 - \alpha)(1 - \beta) + \alpha\beta] \qquad \text{(C.70)}$$

$$P(A = 0, B = 1) = \tfrac{1}{2}[(1 - \alpha)\beta + \alpha(1 - \beta)] \qquad \text{(C.71)}$$

$$P(A = 1, B = 0) = \tfrac{1}{2}[\alpha(1 - \beta) + (1 - \alpha)\beta]. \qquad \text{(C.72)}$$

If $\gamma < 1/2$ make $\beta = 1 - \alpha$ and $\alpha = \frac{1 \pm \sqrt{1 - 2\gamma}}{2}$. If $\gamma \geq 1/2$, let $\beta = \alpha$ and $\alpha = \frac{1 \pm \sqrt{2\gamma - 1}}{2}$. It is just a matter of algebra to verify that $P(A, B)$ results in the intended distribution. $\qquad \square$

### C.1.2 Proof of Theorem 2

**Theorem 2.** *Let $\mathbf{Y}, \mathbf{X}, \mathbf{W} \subset \mathbf{V}$, $\mathbf{W} \cap \mathbf{Y} = \emptyset$, $\sigma_{\mathbf{X}}$ be any intervention, and $\mathcal{G}_{\sigma_{\mathbf{X}}}$ the corresponding interventional causal graph. Then, the effect of $\sigma_{\mathbf{X}}$ on $\mathbf{Y}$ conditioned on $\mathbf{W}$ is given by*

$$P(\mathbf{y}|\mathbf{w}; \sigma_{\mathbf{X}}) = P(\mathbf{y}|\mathbf{w_y}; \sigma_{\mathbf{X}}, \sigma_{\mathbf{W}_{\bar{\mathbf{y}}}} = \mathbf{w}_{\bar{\mathbf{y}}}) = \sum_{\mathbf{a} \backslash (\mathbf{y} \cup \mathbf{w_y})} Q[\mathbf{A}; \sigma_{\mathbf{X}}] \Big/ \sum_{\mathbf{a} \backslash \mathbf{w_y}} Q[\mathbf{A}; \sigma_{\mathbf{X}}], \quad \text{(10)}$$

*where $\mathbf{W_y} \subseteq \mathbf{W}$ is the set of variables in $\mathbf{W}$ connected to any $Y \in \mathbf{Y}$ by any path (regardless of the directionality) in $\mathcal{G}_{\sigma_{\mathbf{X}}[\mathbf{D}]_{\underline{\mathbf{W}}}}$, with $\mathbf{D} = An(\mathbf{Y} \cup \mathbf{W})_{\mathcal{G}_{\sigma_{\mathbf{X}}}}$, $\mathbf{W_{\overline{y}}} = \mathbf{W} \setminus \mathbf{W_y}$, and $\mathbf{A} = An(\mathbf{Y} \cup \mathbf{W_y})_{\mathcal{G}_{\sigma_{\mathbf{X}}\underline{\mathbf{W}}}}$. Furthermore, this effect is transportable from $\langle \mathcal{G}^{\Delta}, \mathbb{Z} \rangle$ iff $Q[\mathbf{A}; \sigma_{\mathbf{X}}]$ is transportable from $\langle \mathcal{G}^{\Delta}, \mathbb{Z} \rangle$.*

*Proof.* First we argue that $(\mathbf{Y} \perp\!\!\!\perp \mathbf{W_{\overline{y}}} \mid \mathbf{W_y})$ in $\mathcal{G}_{\sigma_{\mathbf{X}}\underline{\mathbf{W_{\overline{y}}}}}$ and $\mathcal{G}_{\sigma_{\mathbf{X}}\overline{\mathbf{W_{\overline{y}}}}\underline{\mathbf{W_{\overline{y}}}}}$. The separation in the latter graph is obvious since we are cutting both the incoming and outgoing edges to $\mathbf{W_{\overline{y}}}$. For the first graph suppose the separation does not hold, then let $\overline{q}$ be the path between $Y \in \mathbf{Y}$ and $W \in \mathbf{W_{\overline{y}}}$ that is active in $\mathcal{G}_{\sigma_{\mathbf{X}}\underline{\mathbf{W_{\overline{y}}}}}$ given $\mathbf{W_y}$. Path $\overline{q}$ must not have any edge going out of $\mathbf{W_y}$ else it would be blocked by conditioning on that set. Then, $\overline{q}$ exists in $\mathcal{G}_{\sigma_{\mathbf{X}}\underline{\mathbf{W}}}$, but this would make $W$, by definition, part of $\mathbf{W_y}$, a contradiction. Since the stated separation holds we can use rule 2 of $\sigma$-calculus to infer

$$P(\mathbf{y} \mid \mathbf{w}; \sigma_{\mathbf{X}}) = P(\mathbf{y} \mid \mathbf{w}; \sigma_{\mathbf{X}}, \sigma_{\mathbf{W_{\overline{y}}}} = \mathbf{w_{\overline{y}}}). \tag{C.73}$$

By definition of conditional probability:

$$P(\mathbf{y} \mid \mathbf{w}; \sigma_{\mathbf{X}}) = P(\mathbf{y} \mid \mathbf{w_y}, \mathbf{w_{\overline{y}}}; \sigma_{\mathbf{X}}, \sigma_{\mathbf{W_{\overline{y}}}} = \mathbf{w_{\overline{y}}}) \tag{C.74}$$

$$= \frac{P(\mathbf{y}, \mathbf{w_{\overline{y}}} \mid \mathbf{w_y}; \sigma_{\mathbf{X}}, \sigma_{\mathbf{W_{\overline{y}}}} = \mathbf{w_{\overline{y}}})}{P(\mathbf{w_{\overline{y}}} \mid \mathbf{w_y}; \sigma_{\mathbf{X}}, \sigma_{\mathbf{W_{\overline{y}}}} = \mathbf{w_{\overline{y}}})} \tag{C.75}$$

$$= \frac{P(\mathbf{y} \mid \mathbf{w_y}; \sigma_{\mathbf{X}}, \sigma_{\mathbf{W_{\overline{y}}}} = \mathbf{w_{\overline{y}}}) P(\mathbf{w_{\overline{y}}} \mid \mathbf{y}, \mathbf{w_y}; \sigma_{\mathbf{X}}, \sigma_{\mathbf{W_{\overline{y}}}} = \mathbf{w_{\overline{y}}})}{P(\mathbf{w_{\overline{y}}} \mid \mathbf{w_y}; \sigma_{\mathbf{X}}, \sigma_{\mathbf{W_{\overline{y}}}} = \mathbf{w_{\overline{y}}})} \tag{C.76}$$

$$= P(\mathbf{y} \mid \mathbf{w_y}; \sigma_{\mathbf{X}}, \sigma_{\mathbf{W_{\overline{y}}}} = \mathbf{w_{\overline{y}}}). \tag{C.77}$$

The second factor in the numerator and the denominator in Eq. (C.76) are equal to 1 because of the intervention $\sigma_{\mathbf{W_{\overline{y}}}} = \mathbf{w_{\overline{y}}}$. This proves the first part of the equality in Eq. (10).

From [38] we have that for any disjoint sets $\mathbf{S}, \mathbf{T} \subseteq \mathbf{V}$, $P(\mathbf{s} \mid do(\mathbf{t})) = \sum_{\mathbf{d} \setminus \mathbf{s}} Q[\mathbf{D}]$, where $\mathbf{D} = An(\mathbf{S})_{\mathcal{G}_{[\mathbf{V} \setminus \mathbf{T}]}}$. It is easy to verify that the ancestors of $\mathbf{Y} \cup \mathbf{W_y}$ in $\mathcal{G}_{\sigma_{\mathbf{X}}\underline{\mathbf{W}}}$, $\mathbf{A}$, are the same as in $\mathcal{G}_{\sigma_{\mathbf{X}}[\mathbf{V} \setminus \mathbf{W_{\overline{y}}}]}$, then keeping $\sigma_{\mathbf{X}}$ fixed we could write

$$P(\mathbf{y}, \mathbf{w_y}; \sigma_{\mathbf{X}}, \sigma_{\mathbf{W_{\overline{y}}}} = \mathbf{w_{\overline{y}}}) = \sum_{\mathbf{a} \setminus (\mathbf{y} \cup \mathbf{w_{\overline{y}}})} Q[\mathbf{A}; \sigma_{\mathbf{X}}], \tag{C.78}$$

and then

$$P(\mathbf{y} \mid \mathbf{w}; \sigma_{\mathbf{X}}) = \frac{\sum_{\mathbf{a} \setminus (\mathbf{y} \cup \mathbf{w})} Q[\mathbf{A}; \sigma_{\mathbf{X}}]}{\sum_{\mathbf{a} \setminus \mathbf{w}} Q[\mathbf{A}; \sigma_{\mathbf{X}}]}. \tag{C.79}$$

which proves the second equality in Eq. (10).

Let $\mathcal{G}^{\Delta'}$ be the same as $\mathcal{G}^{\Delta}$ after removing all edges out of $\mathbf{W_y}$ and any edge out of $\mathbf{Y}$ does is not part of a directed path between $\mathbf{Y}$ and $\mathbf{W_y}$. $\mathcal{G}^{\Delta'}$ and $\mathcal{G}^{\Delta}$ have the same C-component structure and all variables in $\mathbf{A}$ are still ancestors of $\mathbf{Y} \cup \mathbf{W}$; therefore and by the same reasoning, $Q[\mathbf{A}; \sigma_{\mathbf{X}}]$ is not transportable from $\langle \mathcal{G}^{\Delta'}, \mathbb{Z} \rangle$ either. Without loss of generality let

$\mathcal{M}^{(i)} = \{\mathcal{M}^{i,k}\}_{\pi^k \in \Pi}$, $i = 1, 2$, be sets of models witnessing the non-transportability of $Q[\mathbf{A}; \sigma_{\mathbf{X}}]$ from $\langle \mathcal{G}^{\Delta'}, \mathbb{Z} \rangle$. Since every variable in $\mathbf{A} \setminus (\mathbf{Y} \cup \mathbf{W_y})$ has a children in $\mathbf{A}$, we can apply Lemma 4 in a topological order over $\mathbf{A} \setminus (\mathbf{Y} \cup \mathbf{W_y})$ and conclude that $Q[\mathbf{A}; \sigma_{\mathbf{X}}]$ transportable if and only if $P(\mathbf{y}, \mathbf{w_y}; \sigma_{\mathbf{X}}, \sigma_{\mathbf{W_{\overline{y}}}} = \mathbf{w_{\overline{y}}}) = \sum_{\mathbf{a} \setminus (\mathbf{y} \cup \mathbf{w_y})} Q[\mathbf{A}; \sigma_{\mathbf{X}}]$ is transportable.

For simplicity, let

$$\rho(\mathbf{y}, \mathbf{w}) = P(\mathbf{y}, \mathbf{w_y}; \sigma_{\mathbf{X}}, \sigma_{\mathbf{W_{\overline{y}}}} = \mathbf{w_{\overline{y}}}) = \sum_{\mathbf{a} \setminus (\mathbf{y} \cup \mathbf{w_{\overline{y}}})} Q[\mathbf{A}; \sigma_{\mathbf{X}}], \tag{C.80}$$

then

$$P(\mathbf{y} \mid \mathbf{w}; \sigma_{\mathbf{X}}) = \frac{\rho(\mathbf{y}, \mathbf{w})}{\sum_{\mathbf{y}} \rho(\mathbf{y}, \mathbf{w})}. \tag{C.81}$$

If $\rho(\mathbf{w}) = \sum_{\mathbf{y}} \rho(\mathbf{y}, \mathbf{w})$ is transportable then $P(\mathbf{y} \mid \mathbf{w}; \sigma_{\mathbf{X}})$ must be non-transportable, else

$$\rho(\mathbf{y}, \mathbf{w}) = P(\mathbf{y} \mid \mathbf{w}; \sigma_{\mathbf{X}}) \rho(\mathbf{w}), \tag{C.82}$$

contradicting the assumption that the $\rho(\mathbf{y}, \mathbf{w})$ is not transportable. Therefore, we can further assume $\rho(\mathbf{w})$ is not transportable, for the rest of the argument.

Let $(\mathbf{y}', \mathbf{w}')$ be an assignment such that $\rho(\mathbf{y}', \mathbf{w}')$ among $\mathcal{M}^{(1)}$ and $\mathcal{M}^{(2)}$, then let

| $\mathbf{W}$ | $\mathbf{Y}$ | $\rho^{(1)}(\mathbf{W}, \mathbf{Y})$ | $\rho^{(2)}(\mathbf{W}, \mathbf{Y})$ |
|---|---|---|---|
| $\mathbf{w}'$ | $\mathbf{y}'$ | $a$ | $b$ |
| $\mathbf{w}'$ | $\neq \mathbf{y}'$ | $c$ | $d$ |
| $\neq \mathbf{w}'$ | $\mathbf{y}'$ | $e$ | $f$ |
| $\neq \mathbf{w}'$ | $\neq \mathbf{y}'$ | $g$ | $h$ |

Due to the non-transportability of $\rho(\mathbf{y}, \mathbf{w})$ we have $a \neq b$ and without loss of generality we can assume $a > b$. Similarly, due to the non-transportability of $\rho(\mathbf{w})$ we have $a + c \neq b + d$. For $\mathcal{M}^{(1)}$ and $\mathcal{M}^{(2)}$:

$$\rho^{(1)}(\mathbf{y}' \mid \mathbf{w}') = \frac{a}{a + c} \tag{C.83}$$

$$\rho^{(2)}(\mathbf{y}' \mid \mathbf{w}') = \frac{b}{b + d}. \tag{C.84}$$

These probabilities are equal if and only if $ad = bc$. Hence, if they are not equal we are done because $\rho^{(1)}(\mathbf{y}' \mid \mathbf{w}') \neq \rho^{(2)}(\mathbf{y}' \mid \mathbf{w}')$. If they are equal, let $W \in \mathbf{W_y}$ be such that there exists a path $\bar{p}$ between it and $Y \in \mathbf{Y}$ in $\mathcal{G}^{\Delta'}$ that does not contain any $\mathbf{W} \setminus \{W\}$. Such $\bar{p}$ exists because by assumption every element in $\mathbf{W_y}$ is connected to some element of $\mathbf{Y}$ in $\mathcal{G}_{\sigma_{\mathbf{x}}[\mathbf{D}]\underline{\mathbf{W}}}$ (which is a subgraph of $\mathcal{G}^{\Delta'}$), so $W$ could be just the closest to some $Y$.

Add a bit to every variable in $\bar{p}$ and denote them with subscript $p$. Define independent functions for the bits which we will parametrize later.

We define two new models $\mathcal{M}^{(1)'}$ and $\mathcal{M}^{(2)'}$, based on $\mathcal{M}^{(1)}$ and $\mathcal{M}^{(2)}$. For every variable in $\bar{p}$ except for $W$ and $Y$, append the corresponding extra bit defined in $\bar{p}$ with the original variables in $\mathcal{M}^{(1)}$ and $\mathcal{M}^{(2)}$. Rename $W$ and $Y$ as $\widetilde{W}, \widetilde{Y}$ and make them unobservable, then define $W$ in the new models with the functions:

$$f_w' = \begin{cases} w' & \text{if } W_p = 1 \\ \widetilde{W} & \text{otherwise,} \end{cases} \tag{C.85}$$

where $W_p$ is unobservable too and $w'$ is the assignment to $W$ consistent with $\mathbf{w}'$ for which the query disagrees in $\mathcal{M}^{(1)}$ and $\mathcal{M}^{(2)}$.

Analogously, define

$$f_y' = \begin{cases} y' & \text{if } Y_p = 1 \\ \widetilde{Y} & \text{otherwise.} \end{cases} \tag{C.86}$$

The path $\bar{p}$ must have a common ancestor to $W$ and $Y$. Such ancestor could be observable or unobservable. That is, either there exists $Z \in \bar{p} \cap An(W) \cap An(Y)$ (possibly $Y$ itself) or there exists $Z_1, Z_2 \in \bar{p}$ with $Z_1 \in An(W), Z_2 \in An(Y)$ and there is an edge $Z_1 \leftarrow\!-\!-\!-\!\rightarrow Z_2$ in $\bar{p}$. For the parametrization of the extra bits in $\bar{p}$ define a new unobservable $U$ and let $Z_p = U$ if the common ancestor is observable or let $U$ be the unobservable parent of $Z_{1p}$ and $Z_{2p}$ in the second. Notice that $Z_1$ and $Z_2$ may be equal to $W$ and $Y$ themselves.

Using lemma 6 we will parametrize $\bar{p}$ such that

| $W_0$ | $Y_0$ | $P(W_0, Y_0)$ |
|---|---|---|
| 1 | 1 | $\frac{1}{2}\gamma$ |
| 1 | 0 | $\frac{1}{2}(1 - \gamma)$ |
| 0 | 1 | $\frac{1}{2}(1 - \gamma)$ |
| 0 | 0 | $\frac{1}{2}\gamma$ |

for some $\gamma \in (0,1)$ that we will pick later.

**Claim 1** (Disagreement on the query). $\mathcal{M}^{(1)'}$ and $\mathcal{M}^{(2)'}$ disagree on the query for any $\gamma$ such that $c - d \neq [(a + c + 1)h - (b + d + 1)g](1 - \gamma)$.

*Proof.* For $\mathcal{M}^{(1)'}$ and $\mathcal{M}^{(2)'}$ we have

$$\rho'(\mathbf{w}', \mathbf{y}') = \sum_{w_p, y_p, \widetilde{w}, \widetilde{y}} \rho'(\mathbf{w}', \mathbf{y}', \widetilde{w}, \widetilde{y} \mid w_p, y_p) P(w_p, y_p). \tag{C.87}$$

Going over each possible combination of $W_p$ and $Y_p$ first we get

$$\rho'(\mathbf{w}', \mathbf{y}') = \rho(\mathbf{W} = \mathbf{w}', \mathbf{Y} = \mathbf{y}') P(W_p = 0, Y_p = 0) \tag{C.88}$$
$$+ \rho(\mathbf{Y} = \mathbf{y}') P(W_p = 1, Y_p = 0) \tag{C.89}$$
$$+ \rho(\mathbf{W} = \mathbf{w}') P(W_p = 0, Y_p = 1) \tag{C.90}$$
$$+ P(W_p = 1, Y_p = 1). \tag{C.91}$$

Similarly,

$$\rho'(\mathbf{w}') = \rho(\mathbf{W} = \mathbf{w}') P(W_p = 0) + P(W_p = 1). \tag{C.92}$$

For $\mathcal{M}^{(1)'}$

$$\rho^{(1)'}(\mathbf{y}' \mid \mathbf{w}') = \frac{\frac{1}{2}a\gamma + \frac{1}{2}(a+e)(1-\gamma) + \frac{1}{2}(a+c)(1-\gamma) + \frac{1}{2}\gamma}{\frac{1}{2}(a+c) + \frac{1}{2}} \tag{C.93}$$

$$= \frac{a\gamma + (2a + c + e)(1-\gamma) + \gamma}{a + c + 1} \tag{C.94}$$

$$= \frac{a - (a + c + e)(\gamma - 1) + \gamma}{a + c + 1} \tag{C.95}$$

$$= \frac{a + (a + c + e)(1 - \gamma) - (1 - \gamma) + 1}{a + c + 1} \tag{C.96}$$

$$= \frac{a - (1 - (a + c + e))(1 - \gamma) + 1}{a + c + 1} \tag{C.97}$$

$$= \frac{a - g(1 - \gamma) + 1}{a + c + 1} \tag{C.98}$$

Analogously for $\mathcal{M}^{(2)'}$:

$$\rho^{(2)'}(\mathbf{y}' \mid \mathbf{w}') = \frac{b - h(1 - \gamma) + 1}{b + d + 1} \tag{C.99}$$

Those two are equal if and only if

$$ab - bg(1 - \gamma) + b + ad - dg(1 - \gamma) + d + a - g(1 - \gamma) + 1$$
$$= ab - ah(1 - \gamma) + a + bc - ch(1 - \gamma) + c + b - h(1 - \gamma) + 1 \tag{C.100}$$
$$\iff$$
$$-bg(1 - \gamma) + ad - dg(1 - \gamma) + d + -g(1 - \gamma)$$
$$= -ah(1 - \gamma) + bc - ch(1 - \gamma) + c + -h(1 - \gamma) \tag{C.101}$$

Recall that we have $ad = bc$, which also implies that $c \neq d$, else $a = b$ which is a contradiction. Then, the condition for equality can be further simplified as

$$c - d = [(a + c + 1)h - (b + d + 1)g](1 - \gamma). \tag{C.102}$$

The left hand side is non-zero, all $a, b, c, d, g$ and $h$ are fixed, and $\gamma$ is a free parameter. Therefore, as long as we pick a $\gamma$ such that the equality doesn't hold, we get that $\rho(\mathbf{y}' \mid \mathbf{w}') = P(\mathbf{y}' \mid \mathbf{w}'; \sigma_{\mathbf{X}})$ does not match in $\mathcal{M}^{(1)'}$ and $\mathcal{M}^{(2)'}$.

$\square$

**Claim 2** (Agreement on any agreed distribution). *Let $\mathbf{Z} \subset \mathbf{V}$ be any subset of observable variables and let $\sigma_{\mathbf{Z}} \in \mathbb{Z}^k \in \mathcal{Z}$. If $\mathcal{M}^{(1)}$ and $\mathcal{M}^{(2)}$ agree on $P^k(\mathbf{V}; \sigma_{\mathbf{Z}})$, then $\mathcal{M}^{(1)'}$ and $\mathcal{M}^{(2)'}$ also agree on $P^k(\mathbf{V}; \sigma_{\mathbf{Z}})$.*

*Proof.* For simplicity we omit the superscript $k$ for the domain, which is fixed with $\sigma_{\mathbf{Z}}$. The superscript $(1)$ and $(2)$ indicate to which of the sets of models under consideration the expression refers to.

Let $\mathbf{C}_1, \mathbf{C}_2, \ldots$ be the C-components of $\mathcal{G}_{\sigma_{\mathbf{Z}}}$. By assumption we have $Q^{(1)}[\mathbf{V}; \sigma_{\mathbf{Z}}] = Q^{(2)}[\mathbf{V}; \sigma_{\mathbf{Z}}]$, and since any $Q[\mathbf{C}_j; \sigma_{\mathbf{Z}}]$ is identifiable from $Q[\mathbf{V}; \sigma_{\mathbf{Z}}]$, we have $Q^{(1)}[\mathbf{C}_j; \sigma_{\mathbf{Z}}] = Q^{(2)}[\mathbf{C}_j; \sigma_{\mathbf{Z}}]$ for any $\mathbf{C}_j$.

$\mathcal{M}^{(k)'}$ is identical to $\mathcal{M}^{(k)}$, $k = 1, 2$, except for the functions of the observables in the path $\bar{p}$. For any variable $T$ not in $\bar{p}$, but with a parent on it, the function $f_T$ remains the same and it simply ignores the extra bit that its parent has in $\mathcal{M}^{(k)'}$.

Let $\mathbf{C}_j$ be a C-component containing some set of variables $\mathbf{R}$ in $\bar{p}$ different to $W$ and $Y$ (the endpoints of $\bar{p}$). First, by definition

$$Q^{(k)'}[\mathbf{C}_j; \sigma_{\mathbf{Z}}](\mathbf{v}) = \sum_{\mathbf{u}(\mathbf{C}_j)} \prod_{V_i \in \mathbf{C}_j} P^{(k)'}(v_i \mid \mathbf{pa}_i, \mathbf{u}_i; \sigma_{\mathbf{X}}) P^{(k)'}(\mathbf{u}(\mathbf{C}_j)). \tag{C.103}$$

For any $S \notin \bar{p}$, $R \in \mathbf{R}$, $W$ and $Y$ that could be in $\mathbf{C}_j$ in $\mathcal{M}^{k'}$, their corresponding factors in the previous expression can be re-written in terms of probabilities of $\mathcal{M}^k$, as follows:

$$P^{(k)'}(s \mid \mathbf{pa}_s, u_s; \sigma_{\mathbf{X}}) = P^{(k)}(s \mid \mathbf{pa}_s, u_s; \sigma_{\mathbf{X}}), \tag{C.104}$$

$$P^{(k)'}(r \mid \mathbf{pa}_r, u_r; \sigma_{\mathbf{X}}) = P^{(k)}(r \mid \mathbf{pa}_r, u_r; \sigma_{\mathbf{X}}) P(r_p \mid (\mathbf{pa}_r)_p), \tag{C.105}$$

$$P^{(k)'}(y \mid \mathbf{pa}_y, u_y; \sigma_{\mathbf{X}}) = P^{(k)}(y \mid \mathbf{pa}_y, u_y; \sigma_{\mathbf{X}}) P(Y_p = 0 \mid (\mathbf{pa}_y)_p)$$
$$+ 1[y = y'] P(Y_p = 1 \mid (\mathbf{pa}_y)_p)), \tag{C.106}$$

$$P^{(k)'}(w \mid \mathbf{pa}_w, u_w; \sigma_{\mathbf{X}}) = P^{(k)}(w \mid \mathbf{pa}_w, u_w; \sigma_{\mathbf{X}}) P(W_p = 0 \mid (\mathbf{pa}_w)_p)$$
$$+ 1[w = w'] P(W_p = 1 \mid (\mathbf{pa}_w)_p)). \tag{C.107}$$

It follows that

$$Q^{(k)'}[\mathbf{C}_j; \sigma_{\mathbf{Z}}](\mathbf{v}) = \left( \prod_{R \in \mathbf{R}} P(r_p \mid (\mathbf{pa}_r)_p) \right) \Big[$$
$$Q^{(k)}[\mathbf{C}_j; \sigma_{\mathbf{Z}}](\mathbf{v}) P(Y_p = 0 \mid (\mathbf{pa}_y)_p) P(W_p = 0 \mid (\mathbf{pa}_w)_p)$$
$$+ Q^{(k)}[\mathbf{C}_j \setminus \{Y\}; \sigma_{\mathbf{Z}}](\mathbf{v}) P(Y_p = 1 \mid (\mathbf{pa}_y)_p) P(W_p = 0 \mid (\mathbf{pa}_w)_p) 1[y = y'] \tag{C.108}$$
$$+ Q^{(k)}[\mathbf{C}_j \setminus \{W\}; \sigma_{\mathbf{Z}}](\mathbf{v}) P(Y_p = 0 \mid (\mathbf{pa}_y)_p) P(W_p = 1 \mid (\mathbf{pa}_w)_p) 1[w = w']$$
$$+ Q^{(k)}[\mathbf{C}_j \setminus \{W, Y\}; \sigma_{\mathbf{Z}}](\mathbf{v}) P(Y_p = 1 \mid (\mathbf{pa}_y)_p) P(W_p = 1 \mid (\mathbf{pa}_w)_p) 1[w = w', y = y']$$
$$\Big].$$

Since $W$ and $Y$ have no descendants in $\mathcal{G}$,

$$Q^{(k)}[\mathbf{C}_j \setminus \{W, Y\}; \sigma_{\mathbf{Z}}](\mathbf{v}) = \sum_{w, y} Q^{(k)}[\mathbf{C}_j; \sigma_{\mathbf{Z}}](\mathbf{v}) \tag{C.109}$$

$$Q^{(k)}[\mathbf{C}_j \setminus \{W\}; \sigma_{\mathbf{Z}}](\mathbf{v}) = \sum_{w} Q^{(k)}[\mathbf{C}_j; \sigma_{\mathbf{Z}}](\mathbf{v}) \tag{C.110}$$

$$Q^{(k)}[\mathbf{C}_j \setminus \{Y\}; \sigma_{\mathbf{Z}}](\mathbf{v}) = \sum_{y} Q^{(k)}[\mathbf{C}_j; \sigma_{\mathbf{Z}}](\mathbf{v}), \tag{C.111}$$

all match between $\mathcal{M}^{(1)}$ and $\mathcal{M}^{(2)}$. Consequently, every C-factor in the right-hand side of (C.108) is the same in those models, and since every other term is also the same in both $\mathcal{M}^{(1)'}$ and $\mathcal{M}^{(2)'}$, we

conclude that $Q^{(1)'}[\mathbf{C}_j; \sigma_{\mathbf{Z}}](\mathbf{v}) = Q^{(2)'}[\mathbf{C}_j; \sigma_{\mathbf{Z}}](\mathbf{v})$, which in turn implies our claim since

$$P^{(k)'}(\mathbf{v}; \sigma_{\mathbf{Z}}) = \prod_j Q^{(k)'}[\mathbf{C}_j; \sigma_{\mathbf{Z}}](\mathbf{v}). \tag{C.112}$$

$\square$

In summary, $\mathcal{M}^{(1)'}$ and $\mathcal{M}^{(2)'}$ induce $\mathcal{G}^{\boldsymbol{\Delta}}$ and matching $\mathbb{Z}$, yet they differ on the value for $P(\mathbf{y}' \mid \mathbf{w}'; \sigma_{\mathbf{X}})$, proving the non-transportability of the query. $\square$

### C.2 Proof of Completeness of the Algorithm

**Theorem 3.** [$\sigma$-TR *Completeness*] *The effect* $P(\mathbf{y} \mid \mathbf{w}; \sigma_{\mathbf{X}})$ *is transportable from* $\mathbb{Z}$ *in* $\mathcal{G}^{\boldsymbol{\Delta}}$ *if and only if the algorithm* $\sigma$-TR *(Alg. 1) outputs an estimand for it.*

*Proof.* $\sigma$-TR fails when there exists some C-component $\mathbf{A}_i$ of $\mathcal{G}_{\sigma_{\mathbf{X}}[\mathbf{A}]}$, a C-component $\mathbf{C}_i$ of $\mathcal{G}^{\boldsymbol{\Delta}}{}_{[An(\mathbf{A}_i)]}$, and for every $\sigma_{\mathbf{Z}} \in \mathbb{Z}^k \in \mathbb{Z}$ at least one of the following three conditions occur:

(i) $\mathbf{A}_i \cap \Delta^k \neq \emptyset$, that is, at least one variable in $\mathbf{A}_i$ has a different mechanism in $\pi^k$, or
(ii) $\mathbf{A}_i \cap \mathbf{Z} \neq \emptyset$, meaning at least one of the variables in $\mathbf{A}_i$ have been intervened by $\sigma_{\mathbf{Z}}$ in $\pi^k$, or
(iii) there exists some $\mathbf{T}_i$ s.t. $\mathbf{A}_i \subset \mathbf{T}_i \subseteq \mathbf{C}_i$, $\mathcal{G}_{\sigma_{\mathbf{Z}}[\mathbf{T}_i]}$ has a single C-component.

Notice that IDENTIFY is called with $\mathbf{B}_i$ which is a C-component in $\mathcal{G}_{\sigma_{\mathbf{Z}}}$. IDENTIFY (in line 4) fails to obtain $Q^*[\mathbf{A}_i] = Q^k[\mathbf{A}_i; \sigma_{\mathbf{Z}}]$ from $Q^k[\mathbf{B}_i; \sigma_{\mathbf{Z}}]$ in $\mathcal{G}_{\sigma_{\mathbf{Z}}}$ only if there exists some $\mathbf{T}_i$ s.t. $\mathbf{A}_i \subset \mathbf{T}_i \subseteq \mathbf{B}_i$ [17, Thm. 3]. Let $\mathbf{B}_i' \subseteq \mathbf{B}_i$ be the variables in $\mathbf{B}_i$ that are ancestors of $\mathbf{A}_i$ in $\mathcal{G}_{\sigma_{\mathbf{Z}}[\mathbf{B}_i]}$. Since the first step identify IDENTIFY takes is to sum out the variables in $\mathbf{B}_i \setminus \mathbf{B}_i'$, $\mathbf{T}_i \subseteq \mathbf{B}_i'$.

Furthermore, we show that $\mathbf{T}_i \subseteq \mathbf{B}_i' \subseteq \mathbf{C}_i$ using an argument by contradiction. Suppose there exists some $B \in \mathbf{B}_i' \setminus \mathbf{C}_i$, then $B$ is not an ancestor of $\mathbf{A}_i$ in $\mathcal{G}$ because $\mathbf{C}_i$ contains all such ancestors that are in the same C-component as $\mathbf{A}_i$ in $\mathcal{G}$. Since $\mathcal{G}_{\sigma_{\mathbf{Z}}}$ could have only less bidirected arrows than $\mathcal{G}$, $B$ must be in the same C-component as $\mathbf{A}_i$ in $\mathcal{G}$. Then, for $B$ to be an ancestor of $\mathbf{A}_i$ only after intervention $\sigma_{\mathbf{Z}}$, there must be some $Z \in \mathbf{Z}$ such that $\sigma_{\mathbf{Z}}$ added an edge $B_1 \to Z$, $B_1 \in \mathbf{B}_i'$, but in any intervened $Z$ forms its own C-component in $\mathcal{G}_{\mathbf{Z}}$ and cannot be part of $\mathbf{B}_i$, a contradiction.

By lemma 10 we have that if the conditions just described are satisfied, there exists a subgraph of $\mathbf{C}_i$ that is an s-Thicket for $P^*(\mathbf{a}_i \mid do(\mathbf{v} \setminus \mathbf{a}_i)) = Q^*[\mathbf{A}_i] = Q^*[\mathbf{A}_i; \sigma_{\mathbf{X}}{=}\mathbf{x}]$.

This implies the existence of two sets of models $\mathcal{M}^{(i)} = \{\mathcal{M}^{i,k}\}_{\pi^k \in \Pi}$, $i = 1, 2$, such that for every $\sigma_{\mathbf{Z}} \in \mathbb{Z}^k \in \mathbb{Z}$ the corresponding models agree on $P^k(\mathbf{v}|do(\mathbf{z}))$, but disagree on $P^*(\mathbf{a}_i \mid do(\mathbf{v} \setminus \mathbf{a}_i))$.

From Eq. (2), and for any conditional or stochastic intervention $\sigma_{\mathbf{Z}}$ the distribution $P(\mathbf{v}; \sigma_{\mathbf{Z}})$ is given by

$$P(\mathbf{v}; \sigma_{\mathbf{Z}}) = \sum_{\mathbf{u}^*} \prod_{\{i|V_i \in \mathbf{Z}\}} P(v_i|\mathbf{pa}_i, \mathbf{u}_i; \sigma_{\mathbf{Z}})P(\mathbf{u}^* \backslash \mathbf{u}; \sigma_{\mathbf{Z}}) \prod_{\{i|V_i \in \mathbf{V} \backslash \mathbf{Z}\}} P(v_i|\mathbf{pa}_i, \mathbf{u}_i)P(\mathbf{u})$$

$$= P(\mathbf{v} \setminus \mathbf{z}|do(\mathbf{z}_j)) \sum_{\mathbf{u}^* \backslash \mathbf{u}} \prod_{\{i|V_i \in \mathbf{Z}\}} P(v_i|\mathbf{pa}_i, \mathbf{u}_i; \sigma_{\mathbf{Z}})P(\mathbf{u}^* \backslash \mathbf{u}; \sigma_{\mathbf{Z}}).$$

Given $\mathcal{M}^{(1)}$ and $\mathcal{M}^{(2)}$, we are free to specify any conditional or stochastic intervention $\sigma_{\mathbf{Z}}$, for any such $\mathbf{Z}$, by setting $P(V_i|\mathbf{Pa}_i, U_i; \sigma_{\mathbf{Z}})$ for every $V_i \in \mathbf{Z}$ in the previous expression, and $P(\mathbf{v}; \sigma_{\mathbf{Z}})$ will be the same as well.

We conclude that $Q^*[\mathbf{A}_i; \sigma_{\mathbf{X}}]$ is not transportable from $\langle \mathcal{G}^{\boldsymbol{\Delta}}, \mathbb{Z} \rangle$. Moreover, $Q[\mathbf{A}; \sigma_{\mathbf{X}}]$ is also not transportable from the same input; were it transportable, Lemma 1 implies that $Q[\mathbf{A}_i; \sigma_{\mathbf{X}}]$ can be obtained from $Q[\mathbf{A}; \sigma_{\mathbf{X}}]$, a contradiction.

Finally, by Thm. 2 we have that $P^*(\mathbf{y}|\mathbf{w}; \sigma_{\mathbf{X}})$ is transportable if and only if $Q^*[\mathbf{A}; \sigma_{\mathbf{X}}]$ is transportable. $\square$

### C.3 Complexity Analysis of the Algorithm

Let $n = |\mathbf{V}|$ and $z = \sum_{\pi^i} |\mathbb{Z}^i|$. Operations in $\sigma$-TR such as computing the set of ancestors or finding the set of C-components in a graph can be done in $O(n^2)$ time. The number of C-components

is at most $n$, hence the total number of times the for-loop in the algorithm could execute, calling *Identify* is $nz$. IDENTIFY (see [38, 16]) recursively reduces the input C-factor at least by a variables each time, and the operations used can be performed in $O(n^2)$; overall it takes $O(n^3)$ time to return an expression or FAIL. Consequently, $\sigma$-TR runs in $O(n^4 z)$.

## C.4 Completeness of Sigma Calculus

First we will show that Lemma 1 follows from $\sigma$-calculus. For simplicity, for any $\mathbf{C}, \mathbf{X} \subseteq \mathbf{V}$ and intervention $\sigma_{\mathbf{X}}^*$ we will write

$$P(\mathbf{c}; \sigma_{\mathbf{X} \cap \mathbf{C}} = \sigma_{\mathbf{X} \cap \mathbf{C}}^*, \sigma_{\mathbf{V} \setminus \mathbf{C}} = (\mathbf{v} \setminus \mathbf{c})) \tag{C.113}$$

simply as

$$P(\mathbf{c}; \sigma_{\mathbf{X(C)}}^*), \tag{C.114}$$

and $Q[\mathbf{C}; \sigma_{\mathbf{X}} = \sigma_{\mathbf{X}}^*]$ as $Q[\mathbf{C}; \sigma_{\mathbf{X}}^*]$.

**Lemma 7** (C-factor – Causal Effect).

$$Q[\mathbf{C}; \sigma_{\mathbf{X}}^*] = P(\mathbf{c}; \sigma_{\mathbf{X(C)}}^*) \tag{C.115}$$

*Proof.* From the model $\mathcal{M}_{\sigma_{\mathbf{X(C)}}}$ we have

$$P(\mathbf{v}; \sigma_{\mathbf{X(C)}}) = \sum_{\mathbf{u}} \prod_i P(v_i \mid \mathbf{pa}_i, \mathbf{u}_i; \sigma_{\mathbf{X(C)}}) P(\mathbf{u}). \tag{C.116}$$

Summing both sides over $\mathbf{V} \setminus \mathbf{C}$

$$P(\mathbf{c}; \sigma_{\mathbf{X(C)}}) = \sum_{\mathbf{u}} \sum_{an(\mathbf{c}) \setminus \mathbf{c}} \prod_{\{i | V_i \in An(\mathbf{C})\}} P(v_i \mid \mathbf{pa}_i, \mathbf{u}_i; \sigma_{\mathbf{X(C)}}) P(\mathbf{u}). \tag{C.117}$$

With $\sigma_{\mathbf{X(C)}}^*$ any variable in $V_i \in An(\mathbf{C}) \setminus \mathbf{C}$ has been fixed to a constant, so each such factor $P(v_i \mid \mathbf{pa}_i, \mathbf{u}_i; \sigma_{\mathbf{X(C)}})$ is 1 when the index of the sum over $An(*C) \setminus \mathbf{C}$ is consistent with $\mathbf{v}$ and 0 otherwise, then

$$P(\mathbf{c}; \sigma_{\mathbf{X(C)}}) = \sum_{\mathbf{u}} \prod_{\{i | V_i \in \mathbf{C}\}} P(v_i \mid \mathbf{pa}_i, \mathbf{u}_i; \sigma_{\mathbf{X(C)}}) P(\mathbf{u}). \tag{C.118}$$

Moreover, any $\mathbf{U}$ that does not appear in $\mathbf{U(C)}$ can be summed out, leaving

$$P(\mathbf{c}; \sigma_{\mathbf{X(C)}}) = \sum_{\mathbf{u(c)}} \prod_{\{i | V_i \in \mathbf{C}\}} P(v_i \mid \mathbf{pa}_i, \mathbf{u}_i; \sigma_{\mathbf{X(C)}}) P(\mathbf{u(c)}). \tag{C.119}$$

For any $V_i \in \mathbf{C}$ the factor $P(v_i \mid \mathbf{pa}_i, \mathbf{u}_i; \sigma_{\mathbf{X(C)}}) = 1$ if and only if $f_i$ in $\mathcal{M}_{\sigma_{\mathbf{X(C)}}^*}$ evaluates to $v_i$. The only such $f_i$ affected by $\sigma_{\mathbf{X(C)}}^* = \sigma_{\mathbf{X}} = \sigma_{\mathbf{X}}^*, \sigma_{\mathbf{V} \setminus \mathbf{C}} = (\mathbf{v} \setminus \mathbf{c})$ are those for $V_i \in \mathbf{X} \cap \mathbf{C}$, and only because of the $\sigma_{\mathbf{X}} = \sigma_{\mathbf{X}}^*$ portion, then

$$P(\mathbf{c}; \sigma_{\mathbf{X(C)}}) = \sum_{\mathbf{u(c)}} \prod_{\{i | V_i \in \mathbf{C}\}} P(v_i \mid \mathbf{pa}_i, \mathbf{u}_i; \sigma_{\mathbf{X}}^*) P(\mathbf{u(c)}), \tag{C.120}$$

which is exactly $Q[\mathbf{C}; \sigma_{\mathbf{X(C)}}^*]$ by definition. $\square$

**Lemma 8** (C-component decomposition). *Let $\mathbf{C}_1, \ldots, \mathbf{C}_l$ be the C-components of $\mathcal{G}_{\sigma_{\mathbf{X}}[\mathbf{C}]}$, let $C_1 < C_2 < C_n$ be any topological order of the variables in $\mathbf{C}$ . Then by $\sigma$-calculus and probability axioms we have*

$$P(\mathbf{c}; \sigma_{\mathbf{X(C)}}^*) = \prod_j P(\mathbf{c}_j; \sigma_{\mathbf{X(C}_j)}^*), \tag{C.121}$$

*where each*

$$P(\mathbf{c}_j; \sigma_{\mathbf{X(C}_j)}^*) = \prod_{\{C_i \in \mathbf{C}_j\}} P(c_i \mid c_1, \ldots, c_{i-1}; \sigma_{\mathbf{X(C)}}^*). \tag{C.122}$$

*Proof.*

$$P(\mathbf{c}; \sigma^*_{\mathbf{X(C)}}) = \prod_i P(c_i \mid c_1, \ldots, c_{i-1}; \sigma^*_{\mathbf{X(C)}}) \tag{C.123}$$

Let $\mathbf{B}_i = \{C_1, \ldots, C_{i-1}\} \setminus \mathbf{C}_j$ (those variables before $C_i$ not in the same C-component as $C_i$). Similarly, let $\mathbf{D}_i = \{C_{i+1}, \ldots, C_l\} \setminus \mathbf{C}_j$.

We have $(C_i \perp\!\!\!\perp \mathbf{D}_i \mid C_1, \ldots, C_{i-1})$ in both $\mathcal{G}_{\sigma_{\mathbf{X(C)}}\overline{\mathbf{D}_i}}$ and $\mathcal{G}_{\sigma_{\mathbf{X}(\{C_1,\ldots,C_i\}\cup\mathbf{C}_j)}\overline{\mathbf{D}_i}}$ because any relevant path would start going out of a variable in $C' \in \mathbf{D}_i$ and is either blocked by a non-observed collider or would entail that $C_i$ goes after $C'$ in the order, which is a contradiction. Then by rule 3 we can exchange $\sigma_{\mathbf{X(C)}}$ with $\sigma_{\mathbf{X}(\{C_1,\ldots,C_i\}\cup\mathbf{C}_j)}$.

Next, $(C_i \perp\!\!\!\perp \mathbf{B}_i \mid \{C_1, \ldots, C_{i-1}\} \setminus \mathbf{B}_i)$ in both $\mathcal{G}_{\sigma_{\mathbf{X}(\{C_1,\ldots,C_i\}\cup\mathbf{C}_j)}\underline{\mathbf{B}_i}}$ and $\mathcal{G}_{\sigma_{\mathbf{X(C}_j)}\underline{\mathbf{B}_i}}$. Any path violating these separations must have an into $C_i$ and an arrow into some $C' \in \mathbf{B}_i$. Since $C_i$ and $C'$ are not in the same C-component, the path must have at least one directed arrow in it; but the variable at the tail of such arrow is either observed (is in the same C-component as $C_i$) or has the outgoing arrows removed (it is in $\mathbf{B}_i$), in any case the path is blocked or non-existent. Then, by rule 2 the intervention can be changed to $\sigma_{\mathbf{X(C}_j)}$ and we have

$$P(c_i \mid c_1, \ldots, c_{i-1}; \sigma^*_{\mathbf{X(C)}}) = P(c_i \mid c_1, \ldots, c_{i-1}; \sigma^*_{\mathbf{X(C}_j)}). \tag{C.124}$$

Under intervention $\sigma^*_{\mathbf{X(C}_j)}$ any variable in $\mathbf{B}_i$ has been fixed to a constant then

$$P(c_i | c_1, \ldots, c_{i-1}; \sigma^*_{\mathbf{X(C}_j)}) = \frac{P(c_i | \{c_1, \ldots, c_{i-1}\} \setminus \mathbf{b}_i; \sigma^*_{\mathbf{X(C}_j)}) P(\mathbf{b}_i | \{c_1, \ldots, c_i\} \setminus \mathbf{b}_i; \sigma^*_{\mathbf{X(C}_j)})}{P(\mathbf{b}_i | \{c_1, \ldots, c_{i-1}\} \setminus \mathbf{b}_i; \sigma^*_{\mathbf{X(C}_j)})} \tag{C.125}$$

$$= P(c_i | \{c_1, \ldots, c_{i-1}\} \setminus \mathbf{b}_i; \sigma^*_{\mathbf{X(C}_j)}), \tag{C.126}$$

because the second factor of the denominator and the denominator are equal to 1. Reorganizing the factors by C-components we get

$$P(\mathbf{c}; \sigma^*_{\mathbf{X(C)}}) = \prod_j \prod_{\{i | C_i \in \mathbf{C}_j\}} P(c_i \mid \{c_1, \ldots, c_{i-1}\} \setminus \mathbf{b}_i; \sigma^*_{\mathbf{X(C}_j)}) \tag{C.127}$$

$$= \prod_j P(\mathbf{c}_j; \sigma^*_{\mathbf{X(C}_j)}), \tag{C.128}$$

which matches Eq. (C.122). Eq. (C.127) together with Eq. (C.126) imply Eq. (C.122). $\qquad\square$

**Lemma 9.** *Each step of* IDENTIFY *follows from σ-calculus.*

*Proof.* As long as the $Q$ and $\mathcal{G}$ in the input to IDENTIFY correspond to the same intervention (i.e., $Q = Q[\mathbf{T}; \sigma_{\mathbf{X}}]$, $\mathcal{G} = \mathcal{G}_{\sigma_{\mathbf{X}}}$, we can keep the context of the intervention implicit.

The input $Q = Q[\mathbf{T}]$ corresponds to $P(\mathbf{t}; \sigma_{\mathbf{X(T)}})$. In line 2 returns $P(\mathbf{c}; \sigma_{\mathbf{X(C)}}) = \sum_{\mathbf{t}\setminus\mathbf{c}} P(\mathbf{t}; \sigma_{\mathbf{X(T)}})$ if $\mathbf{A} = An(\mathbf{C})_{\mathcal{G}_{[\mathbf{T}]}} = \mathbf{C}$, that is, every ancestor of $\mathbf{C}$ in $\mathcal{G}_{[\mathbf{T}]}$ is already in $\mathbf{C}$.

First we argue that $(\mathbf{C} \perp\!\!\!\perp \mathbf{T} \setminus \mathbf{C})$ in $\mathcal{G}_{\sigma_{\mathbf{X(C)}}\overline{\mathbf{T}\setminus\mathbf{C}}}$ and $\mathcal{G}_{\sigma_{\mathbf{X(T)}}\overline{\mathbf{T}\setminus\mathbf{C}}}$. In those graphs all arrows incoming to variables not in $\mathbf{C}$, including $\mathbf{T} \setminus \mathbf{C}$, are cut. Then, any path between some $T \in \mathbf{T} \setminus \mathbf{C}$ and some $C \in \mathbf{C}$ must start with an arrow going out from $T$. If the other end of the edge is not in $\mathbf{C}$ then the path does not exists in the graphs mentioned before. If the edge goes to some variable in $\mathbf{C}$ we have that $T$ is an ancestor of $\mathbf{C}$ in $\mathcal{G}_{[\mathbf{T}]}$ which is assumed not to be the case. Then by rule 3 of σ-calculus:

$$P(\mathbf{c}; \sigma_{\mathbf{X(C)}}) = P(\mathbf{c}; \sigma_{\mathbf{X(T)}}), \tag{C.129}$$

and summing over $\mathbf{T} \setminus \mathbf{C}$:

$$P(\mathbf{c}; \sigma_{\mathbf{X(C)}}) = \sum_{\mathbf{t}\setminus\mathbf{c}} P(\mathbf{t}; \sigma_{\mathbf{X(T)}}), \tag{C.130}$$

as desired.

**Algorithm 2** IDENTIFY($\mathbf{C}, \mathbf{T}, Q, \mathcal{G}$)

---

**Input**: $\mathbf{C} \subseteq \mathbf{T} \subseteq \mathbf{V}$, $Q = Q[\mathbf{T}]$ and graph $\mathcal{G}$. Assuming $\mathcal{G}_{[\mathbf{C}]}$ and $\mathcal{G}_{[\mathbf{T}]}$ are composed of a single c-component.
**Output**: Expression for $Q[\mathbf{C}]$ in terms of $Q$ or Fail.

1: Let $\mathbf{A} \leftarrow An(\mathbf{C})_{\mathcal{G}_{[\mathbf{T}]}}$.
2: **if $\mathbf{A} = \mathbf{C}$ then return** $Q[\mathbf{C}] = \sum_{\mathbf{t} \setminus \mathbf{c}} Q$.
3: **if $\mathbf{A} = \mathbf{T}$ then return** *Fail*.
4: **if $\mathbf{A} = \mathbf{C}$ then**
5:  Let $\mathbf{T}'$ be the C-component containing $\mathbf{C}$ in $\mathcal{G}_{[\mathbf{A}]}$.
6:  Compute $Q[\mathbf{T}']$ from $Q[\mathbf{A}] = \sum_{\mathbf{t} \setminus \mathbf{a}} Q$.
7:  **return** Identify$(\mathbf{C}, \mathbf{T}', Q[\mathbf{T}'], \mathcal{G})$.
8: **end if**

---

In line 6 we follow the same reasoning as above to conclude that $P(\mathbf{a}; \sigma_{\mathbf{X}(\mathbf{A})}) = \sum_{\mathbf{t} \setminus \mathbf{a}} P(\mathbf{t}; \sigma_{\mathbf{X}(\mathbf{T})})$. Then by Lemma 8 we can obtain $P(\mathbf{t}'; \sigma_{\mathbf{X}(\mathbf{T}')})$ from $P(\mathbf{a}; \sigma_{\mathbf{X}(\mathbf{A})})$. Finally, IDENTIFY is called recursively in line 7.

With this all steps have been shown to follow from $\sigma$-calculus and standard probability axioms. $\square$

**Corollary 2.** *[$\sigma$-calculus Completeness] The $\sigma$-calculus together with standard probability axioms is complete for the task of transportability with soft interventions.*

*Proof.* From the first part of the proof of Thm. 2 it is shown with $\sigma$-calculus that

$$P^*(\mathbf{y} \mid \mathbf{w}; \sigma_{\mathbf{X}}) = P^*(\mathbf{y} \mid \mathbf{w}_{\mathbf{y}}; \sigma_{\mathbf{X}}, \sigma_{\mathbf{W}_{\overline{\mathbf{y}}}} = \mathbf{w}_{\overline{\mathbf{y}}}), \qquad (\text{C.131})$$

and by the definition of conditional probability we can write the effect as

$$P^*(\mathbf{y} \mid \mathbf{w}; \sigma_{\mathbf{X}}) = \frac{P^*(\mathbf{y}, \mathbf{w}_{\mathbf{y}}; \sigma_{\mathbf{X}}, \sigma_{\mathbf{W}_{\overline{\mathbf{y}}}} = \mathbf{w}_{\overline{\mathbf{y}}})}{P^*(\mathbf{w}_{\mathbf{y}}; \sigma_{\mathbf{X}}, \sigma_{\mathbf{W}_{\overline{\mathbf{y}}}} = \mathbf{w}_{\overline{\mathbf{y}}})}. \qquad (\text{C.132})$$

From this point we can focus on transporting $P^*(\mathbf{y}, \mathbf{w}_{\mathbf{y}}; \sigma_{\mathbf{X}}, \sigma_{\mathbf{W}_{\overline{\mathbf{y}}}} = \mathbf{w}_{\overline{\mathbf{y}}})$. For simplicity, relabel $\mathbf{Y}$ to represent both $\mathbf{Y} \cup \mathbf{W}_{\mathbf{y}}$ and $\sigma_{\mathbf{X}}^*$ being both $\sigma_{\mathbf{X}}, \sigma_{\mathbf{W}_{\overline{\mathbf{y}}}} = \mathbf{w}_{\overline{\mathbf{y}}}$ and notice that $\mathbf{A} = \mathbf{D} = An(\mathbf{Y} \cup \mathbf{W}_{\mathbf{y}})_{\mathcal{G}_{\sigma_{\mathbf{X}} \sigma_{\mathbf{W}_{\overline{\mathbf{y}}}} = \mathbf{w}_{\overline{\mathbf{y}}}}}$, so that we can continue with the marginal effect of the numerator in mind. Following the same derivation shown in the first part of the proof in Thm. 1 to show Eq. (5) we get

$$P^*(\mathbf{y}; \sigma_{\mathbf{x}} = \sigma_{\mathbf{X}}^*) = \sum_{\mathbf{a} \setminus \mathbf{y}} P^*(\mathbf{a} \setminus \mathbf{x}; \sigma_{\mathbf{X}} = \mathbf{x}) \sum_{X \in \mathbf{A} \cap \mathbf{X}} P^*(x \mid \mathbf{pa}_x; \sigma_{\mathbf{X}} = \sigma_{\mathbf{X}}^*). \qquad (\text{C.133})$$

The sum $\sum_{X \in \mathbf{A} \cap \mathbf{X}} P^*(x \mid \mathbf{pa}_x; \sigma_{\mathbf{X}} = \sigma_{\mathbf{X}}^*)$ is determined by the definition of $\sigma_{\mathbf{X}}^*$ and justifies line 9. Then all that is left is to transport $P^*(\mathbf{a} \setminus \mathbf{x}; \sigma_{\mathbf{X}} = \mathbf{x})$. Since $\mathbf{A} \setminus \mathbf{X} = \mathbf{A} \setminus (\mathbf{X} \cap \mathbf{A})$ and

$$P^*(\mathbf{a} \setminus (\mathbf{x} \cap \mathbf{a}); \sigma_{\mathbf{X}} = \mathbf{x}) = P^*(\mathbf{a} \setminus (\mathbf{x} \cap \mathbf{a}); \sigma_{\mathbf{X} \cap \mathbf{A}} = \mathbf{x} \cap \mathbf{a}), \qquad (\text{C.134})$$

because $\mathbf{A}$ is ancestral in $\mathcal{G}_{\sigma_{\mathbf{X}}} = \mathcal{G}_{\overline{\mathbf{X}}}$ so there is no active path between any $\mathbf{A}$ and $\mathbf{X} \setminus \mathbf{A}$ in $\mathcal{G}_{\sigma_{\mathbf{X} \cap \mathbf{A}} \sigma_{\mathbf{X} \setminus \mathbf{A}} \overline{\mathbf{X} \setminus \mathbf{A}}}$ and $\mathcal{G}_{\sigma_{\mathbf{X} \cap \mathbf{A}} \overline{\mathbf{X} \setminus \mathbf{A}}}$; so this equality is justified by rule 3 and $(\mathbf{A} \perp\!\!\!\perp \mathbf{X} \setminus \mathbf{A})$ in those graphs.

The product in line 11 follows from $\sigma$-calculus as in Lemma 8 with $\mathbf{A}_i$ specified as in line 1.

All that is left is to justify that each $Q^*[\mathbf{A}_i; \sigma_{\mathbf{X}}^*]$ can be derived using $\sigma$-calculus too. To see, notice that $Q^k[\mathbf{B}_i; \sigma_{\mathbf{Z}}] = P^k(\mathbf{b}_i; \sigma_{\mathbf{Z}(\mathbf{B}_i)})$ is obtainable from $P^k(\mathbf{V}; \sigma_{\mathbf{Z}})$ by Lemma 7.

By Lemma 9 we have that if IDENTIFY does not fail to obtain $Q^k[\mathbf{A}_i; \sigma_{\mathbf{Z}}] = P^k(\mathbf{a}_i; \sigma_{\mathbf{Z}(\mathbf{A})})$ from $P^k(\mathbf{b}_i; \sigma_{\mathbf{Z}(\mathbf{B}_i)})$. Due the conditions in line 2 $\mathbf{A}_i$ contains no element in $\mathbf{Z}$, then since $\mathbf{A}_i \cap \mathbf{X}$ is also empty:

$$P^k(\mathbf{a}_i; \sigma_{\mathbf{Z}(\mathbf{A})}) = P^k(\mathbf{a}_i; \sigma_{\mathbf{Z} \cap \mathbf{A}_i}, \sigma_{\mathbf{V} \setminus \mathbf{A}_i} = (\mathbf{v} \setminus \mathbf{a}_i)) \qquad (\text{C.135})$$

$$= P^k(\mathbf{a}_i; \sigma_{\mathbf{V} \setminus \mathbf{A}_i} = (\mathbf{v} \setminus \mathbf{a}_i)) \qquad (\text{C.136})$$

$$= P^k(\mathbf{a}_i; \sigma_{\mathbf{X} \cap \mathbf{A}_i}, \sigma_{\mathbf{V} \setminus \mathbf{A}_i} = (\mathbf{v} \setminus \mathbf{a}_i)) \qquad (\text{C.137})$$

$$= P^k(\mathbf{a}_i; \sigma_{\mathbf{X}(\mathbf{A})}). \qquad (\text{C.138})$$

Finally, we argue that $(\mathbf{A}_i \perp\!\!\!\perp \{S_{V_i} \mid V_i \in \Delta^k\})$ in $\mathcal{G}^{\mathbf{\Delta}}_{\overline{\mathbf{V} \setminus \mathbf{A}_i}}$. Also by condition in line 2 no variable in $\mathbf{A}_i$ is also in $\Delta^k$, so any path between an $S$ variable is cut in this graph. Then

$$P^k(\mathbf{a}_i; \sigma_{\mathbf{X}(\mathbf{A})}) = P^*(\mathbf{a}_i; \sigma_{\mathbf{X}(\mathbf{A})}). \tag{C.139}$$

$\square$

# D   Proofs for Section 4

**Corollary 3.** *Given query* $P^*(\mathbf{y}; \sigma_{\mathbf{X}}=\sigma_{\mathbf{X}}^*)$, *selection diagram* $\mathcal{G}^{\mathbf{\Delta}}$, *and the distribution specified by* $\mathbb{Z}$, *let* $\mathbf{A}$ *be defined as in Thm. 2. Then, the query is not transportable from* $\langle \mathcal{G}^{\mathbf{\Delta}}, \mathbb{Z} \rangle$ *if and only if there exists a C-component* $\mathbf{A}_i$ *of* $\mathcal{G}_{\sigma_{\mathbf{X}}^*[\mathbf{A}]}$ *and a C-component* $\mathbf{C}_i$ *of* $\mathcal{G}^{\mathbf{\Delta}}_{[An(\mathbf{A}_i)]}$ *such that for every* $\sigma_{\mathbf{Z}} \in \mathbb{Z}^k \in \mathbb{Z}$, $\mathbf{A}_i$ *satisfies conditions (i), (ii), or (iii).*

*Proof.* The conditions in the corollary are satisfied if and only if $\sigma$-TR fails, therefore, by Thm. 3 it follows that the query is not transportable from $\langle \mathcal{G}^{\mathbf{\Delta}}, \mathbb{Z} \rangle$. $\square$

**Lemma 10** (s-Thicket relation). *Let* $\mathbf{A}$ *be defined as in Thm. 2 and let* $\mathbf{A}_i$ *be a C-component of* $\mathcal{G}_{\sigma_{\mathbf{X}}[\mathbf{A}]}$. *The condition in Corollary 3 occurs if and only if there exists an s-Thicket* $\mathcal{T}$ *for* $P^*(\mathbf{a}_i \mid do(\mathbf{v} \setminus \mathbf{a}_i))$. *Furthermore, the same* $\mathcal{T}$ *is an s-Thicket for* $P(\mathbf{a} \setminus \mathbf{x} \mid do(\mathbf{x}, an(\mathbf{a}) \setminus \mathbf{a}))$.

*Proof.* Suppose there exists $\mathbf{A}_i$ and $\mathbf{C}_i$ satisfying the conditions in the corollary for each $\sigma_{\mathbf{Z}} \in \mathbb{Z}^k \in \mathbb{Z}$. Recall that according to Thm. 2 the set $\mathbf{A}$ is defined as $An(\mathbf{Y} \cup \mathbf{W_y})_{\mathcal{G}_{\sigma_{\mathbf{X}} \mathbf{W}}}$ and $\mathbf{A}_i \subseteq \mathbf{A}$. Let $\mathcal{T}$ be a minimal subgraph of $\mathcal{G}^{\mathbf{\Delta}}_{[\mathbf{C}_i]}$ such that every edge (directed or bidirected) that can be removed without changing the fact that $\mathbf{S}$ has a single C-component and the ancestral relationships between the variables, have been removed. Then we can verify that $\mathcal{T}$ is an s-Thicket [23, Def. 4] for $P^*(\mathbf{y}, \mathbf{w_y}; do(\mathbf{x}, \mathbf{w_{\overline{y}}}))$ relative to $\mathcal{G}$ and $\mathbb{Z}'$ where all non-atomic interventions are $\mathbb{Z}$ are replaced with atomic ones. To witness, we argue each part of the definition for s-Thicket:

- $\mathcal{T}$ is a minimal subgraph of $\mathcal{G}$ made of a single C-component,

- by conditions (i) and (ii) we have $\Delta^k \cap \mathbf{R} \neq \emptyset$ and $\mathbf{Z} \cap \mathbf{R} \neq \emptyset$ for every $\mathbf{Z}$ s.t. $\sigma_{\mathbf{Z}}=\mathbf{z}$ in $\mathbb{Z}'$.

- Also for every $\mathbf{Z}$, by condition (iii) there exists $\mathbf{T}_i$ such that $\langle \mathbf{T}_i, \mathbf{A}_i \rangle$ is a hedge [23].

- $\mathcal{T}$'s root set $\mathbf{R}$ (the variables in $\mathcal{T}$ without any child) is exactly $\mathbf{A}_i$, then $\mathbf{R} \subseteq An(\mathbf{A}_i)_{\mathcal{G}_{\mathbf{V} \setminus (\mathbf{X} \cup An(\mathbf{A}) \setminus \mathbf{A})}} \subseteq An(\mathbf{A})_{\mathcal{G}_{\mathbf{V} \setminus (\mathbf{X} \cup An(\mathbf{A}) \setminus \mathbf{A})}}$.

- Finally, for every hedgelet in those hedges, $\mathbf{T}_i \setminus \mathbf{A}_i$ has to intersect $\mathbf{X} \cup An(\mathbf{A} \setminus \mathbf{A})$.

$\square$