[Reviews · NeurIPS 2020]

Review 1

Summary and Contributions: This paper presents completeness results for the general transportability of soft interventions in SCMs. Specifically, the authors continue the line of work on general transportability, extending the work done on atomic/do interventions to support more realistic soft/stochastic/policy interventions and showing that the latter can reduce to the former. The authors present an algorithm to determine necessary and sufficient conditions under which the target effect is computable from a mix of observation data from multiple domains. Finally, the authors graphically characterize soft transportability presenting conditions under which one can infer the lack of transportability visually.

Strengths: This paper does an excellent job of motivating the setting and contributions. The notations are well-defined and compact in one place, making it easy to refer back to when needed. The two illustrative examples used in the paper are helpful in understanding concepts.

Weaknesses: As somebody who has studied causality, has worked with the SCM framework and do-calculus, and to some extent has familiarity with soft interventions, I find myself overwhelmed at times. Please see my comment below on clarity. Of course, an experiment with simulated data based on the proposed graphical models would make this paper complete.

Correctness: The motivation and situation in historical context is correct. The claims and methods seem not to have any glaring issues. The depth of technical methodology in the main body seems correct.

Clarity: Overall, yes. The paper is well-written and polished. My only concern regarding the clarity was that the notation (both formulaic and graphical) can be overwhelming at times, even for somebody famililar with the literature. Also, the sudden mentioning of "standard transportability algorithms" in line 181 or "s-Thicket" in line 290 may exude confidence that the authors are familiar with the literature, however, as a reader, this is quite distracting. On the other hand, I understand that the presented work's contribution would not easily fit within the page limits, and thus I am content with the density of the presented material. My suggestion would be to rethink whether some parts are necessary in the main body, or move some of the comments that you don't take the time to fully introduce to footnotes. - fig 2 & 3: the order in which these are presented is confusing and I found myself having to re-read these parts multiple times. Perhaps, lines 127, 128 should go after the description of domains \pi^1, \pi^2 on lines 142-145 and you can merge both figures into (just an idea) a 2-line figure with the first line: Fig 2a, Fig 3a, Fig 3b, and second line: Fig 2b, Fig 2c. This way it's clear what the domains are, it's clear what the intended policy in \pi^* is, and consequently \G^\Delta is immediately inferrable. merging the figures might also save some space, allowing to bring back some of prepositions that were sporadically dropped :)

Relation to Prior Work: To a good extent, yes. - line 44: missing citations on soft-interventions: Eberhardt. ``Causation and intervention'', and Korb+Hope+Nicholson+Axnick. ``Varieties of causal intervention.'' - line 100: this is something I did not know was considered before: your definition of soft interpretability allows for the post-manipulated str. eqn. to depend on NEW parents (both exogenous or endogenous). Prior definitions that I have encountered only say that the influence of pre-manipulation parents may continue to effect the intervened upon node. Perhaps a brief clarification (even in the footnote) may prevent confusion. relatedly, it seems that the authors are using ";" before soft interventions \sigma, in a sense mirroring the notation of hard interventions as the reserved do-operator. it would be good to clarify this also for the majority who are not familiar with soft interventions.

Reproducibility: No

Additional Feedback: Scattered nits and questions: - line 61: why is the distribution of Z in \pi^2 different from that in \pi^*? - line 65: P^*(Y;...) should be P^*(y;...) for consistency with (1) and later formulations - line 72: P^*(y;\sigma_X) should be P^*(y;\sigma_X^*) for consistency - line 108: the subscript of the union writes `X \in \bold{X}`, but it should be `x \in \bold{X}` - line 136: S_v seems to be undefined (I can't seem to find it in Definition 2 or on page 2) - line 151: later --> latter - line 177: is D = d? otherwise D is undefined. - line 268: there are two Z^1s here; the latter should be Z^2 - line 297: while "heterogenous" is clear after reading the paper in full, introducing this term in the conclusion should likely be avoided More: - equation 3: I'm not sure I follow this; where did the W variable go? I understand that the intervention on X removed its direct dependance on W, but can we ignore the bidirectional relation between W and {R,Z}? - fig 2b: I can't seem to find the description of why the intervention \sigma_X introduces the dependence on R. perhaps add a brief statement about this policy? if it helps, you mention this much later in lines 286, 287 but again without context. - fig 4: this should really be corrected and made to look consistent with the earlier figures (in the final version of your paper) ------------------------------------------------------------------------------------ Post-rebuttal comment: I have read the author rebuttal as well as the comments from other reviewers. My review is not changed.


Review 2

Summary and Contributions: In this work the authors present a formal treatment of transportability settings in the context of soft interventions. They develop necessary and sufficient graphical criteria for deciding soft transportability. Furthermore they develop an algorithm to determine if a non-atomic intervention is computable from a combination of distributions available across domains.

Strengths: This work presents a theoretically grounded solution to the problem of transportability of soft interventions. The solution can in the future perhaps guide the development of practical algorithms.

Weaknesses: While the paper starts with a grand introduction and motivation, it quickly reduces to a bunch of results in the form of lemmas and theorems that are not explained or motivated well. What I find most troubling is that all the proofs (every single one of them) have been moved to the appendix. I wouldn't accept lack of space as an excuse for this because the authors include results like lemma 1 (from a paper in the year 2002) in the main paper that are not essential and could easily be moved to the appendix, and can be well explained with an example.

Correctness: This is a purely theoretical results based paper. However, not even a proof sketch has been provided in the main paper. A good practice for such papers is to provide proofs of sufficiency in the main paper and defer the proofs of necessity to the appendix. I have not checked all the proofs in the 28 pages long supplementary materials.

Clarity: The first three pages including the introduction are well-written and they explain & motivate the problem well. Line 84: "as the union of C.." This sentence probably needs to be reworded. Lines 119-124: It would have been clearer if the letters used to denote variables were in some way related to the variable. For instance use C for credit history instead of W. Also I find the second story pertaining to figure 2 confusing. On the other hand the story pertaining to figure 1 was very well put and easily understandable. The quality of writing deteriorates page 4 onwards. Line 139: "while the percentage ought is above"

Relation to Prior Work: I find it hard to believe that [29] is the only related work done by J Robins. It is very likely that the field of epidemiology/public health have more work on this. In fact, figure 1 is a typical problem in these fields.

Reproducibility: Yes

Additional Feedback: 1. Please comment on the part of the review related to prior work. 2. Definition 1 [domain discrepancy]. Note that the term appears only once in the entire paper (i.e. in the definition itself). Also the wording in the definition is sort of confusing. 3. I guess a good understanding of [21] is necessary to understand the surprising element the authors are trying to bring to our attention on page 5 (lines 180-187). Perhaps that part could be written better. 4. Please run spell check. 5. In general, I think there are way too many results crammed into this paper, resulting in the authors having not enough space to clearly explain them. Updating the review The paper lacks clarity and is hard to comprehend without a strong understanding of transportability results. My concern above regarding too many results remain true. Also the authors could have done a better job with regard to citing relevant work.


Review 3

Summary and Contributions: The paper describes a complete algorithm for identifying transportability for soft interventions, as well as a graphical characterization. I acknowledge reading the author's rebuttal.

Strengths: The paper seems theoretically sound and follows up an important line of work, transportability, that allows one to potentially infer the effects of interventions from existing observational and experimental data, generalizing it to the most general type of interventions, soft interventions.

Weaknesses: While the topic is significant for the general ML community, the paper is very technical and written for a very specific audience, so it maybe hard to read for the general NeurIPS attendee. Also, there is no evaluation section, so it might be a bit more difficult to evaluate the practical impact of the work.

Correctness: As far as I know the paper is correct.

Clarity: The paper is clearly written if one is familiar with the transportability literature, but it might be difficult to follow otherwise.

Relation to Prior Work: The paper is well-positioned and its relationship with the related work is clear.

Reproducibility: Yes

Additional Feedback: A few typos: L39: focuses, L157: particular

[Author Response · NeurIPS 2020]

We thank each reviewer for reading our paper and sharing your thoughts and suggestions. We address each review in turn:

**Review #1** We understand your observation regarding the mention of "standard transportability algorithms" and "s-Thicket" which may be distracting for many readers, and share this concern with you. Our motivation to keep such references was to acknowledge relevant work. Still, we do appreciate your suggestion to move these references to footnotes to improve readability.

Also, thank you for pointing out the possible confusion with Figures 2 and 3, and the description of $\pi^1$ and $\pi^2$. We will certainly make the appropriate changes around this section to improve readability.

Regarding your particular questions:

- line 61: The distribution of $Z$ (secondary condition) may be different in $\pi^2$ compared to $\pi^*$, for example, if it represents hypertension and due to the different in average age of the populations, the condition is more prominent in $\pi^2$.

- line 177: $\mathbf{d}$ is an instantiation of the set of variables $\mathbf{D}$.

- In Equation (3) it is not necessary to include $W$. One way to see why is to look at Fig. 2b an notice that $W$ does not have any causal influence on any other variable under $\sigma_X$. Further, under this regime the latent confounding between $W$ and both $R$ and $Z$ turns out to be irrelevant at this point of the derivation.

- In Fig. 2b the influence of $R$ (characteristics of the property) on $X$ is motivated by the description in line 138-139 that goes as "increase the percentage $X$ by offering loaners insurance from default, while the percentage ought is above the regular threshold, for properties within certain locations", where location is a characteristic of the property. We will make this more explicit.

**Review #2**

It seems that the primary concern posed in the review is that the large number of results presented in the form of lemmas, theorems and corollaries without explanation or motivation.

As pointed out, we did not include proofs in the main text, mainly due to space constraints. However, we did intent to explain, at least in terms of statements and implications, each one of the presented results. For subsequent versions of the manuscript, we will try to make such explanations and motivation more prominent and understandable, within the space constraints.

The reviewer calls attention on the possibility of having other results such as lemma 1 not stated but moved to the appendix and explained with an example. It is evident that this is a valid choice, yet in this particular case at least, there is no much space that can be saved. Nevertheless, we could certainly improve the paper with this point in mind, bringing into the main text a proof sketch or main ideas of the proofs for some of the results.

Regarding your specific comments and suggestions:

1. The reference to Pearl and Robins 1995 is motivated by the fact that it is seminal work in dynamic plans literature in the context of causal inference where non-atomic interventions play a crucial role. We are not aware of further work by James Robins with a relevant relationship with the topic of our paper (transportability, multiple domains, experimental conditions) beyond the soft intervention aspect already addressed. Still, we would appreciate it if you would like to share specific references of this or any other author in the epidemiology/public health literature, thank you.

2. Please note that although the term "domain discrepancy" is not used, the symbol $\Delta^{i,j}, \Delta^i$ that entails the concept is widely used in the paper later (e.g., Definition of Selection Diagram, Lemma 3, Alg. 1, and Sec. 4). We were trying to save space leveraging the definition we introduced, instead of having to repeat this relatively long expression, "domain discrepancy."

3. We will certainly look for a better way to convey the point regarding previously proposed transportability algorithms (lines 180-187). Nevertheless, we addressed this point in more detail in section B.3 of the supplemental material.

5. We acknowledge that the density of results in the paper is high. However, we believe that there is also high cohesion between them such that presenting them in parts might not be a better alternative.

**Review #3**

We are glad you find our work sound and belonging to an important line of work, thank you. We agree with your appreciation that the paper requires a certain level of familiarity with the topic to be well understood. Still, we believe this is the case for most papers that introduce new theoretical results in any area in a premier venue such as NeurIPS.

[Meta-Review · NeurIPS 2020]

The paper describes a sound and complete algorithm for identifying responses to soft interventions for transportability problems. The reviewers found the paper interesting and novel, although they found the presentation difficult to follow for an interested member of the NeurIPS community not intimately familiar with details of identification theory. I must second the disappointment expressed in one of the comments of reviewer 2, regarding the somewhat selective "citation discipline" in this paper.